# Adaptive Accelerated Gradient Descent Methods for Convex Optimization

## Abstract

This work proposes A$^2$GD, a novel adaptive accelerated gradient descent method for convex and composite optimization. Smoothness and convexity constants are updated via Lyapunov analysis. Inspired by stability analysis in ODE solvers, the method triggers line search only when accumulated perturbations become positive, thereby reducing gradient evaluations while preserving strong convergence guarantees. By integrating adaptive step size and momentum acceleration, A$^2$GD outperforms existing first-order methods across a range of problem settings.

## 1 Introduction

In this paper, we study the convex optimization problem

$$\min_{x \in \mathbb{R}^d} f(x), \tag{1}$$

where $f$ is $\mu$-strongly convex and $L$-smooth. When $\mu = 0$, we additionally assume $f$ is coercive so that a global minimizer exists. We also consider the composite convex problem

$$\min_{x \in \mathbb{R}^d} f(x) := h(x) + g(x), \tag{2}$$

where $h$ is $L$-smooth and $g$ is convex, possibly non-smooth, with a proximal operator.

First-order methods, which rely only on gradient information, are widely used in machine learning for their efficiency and scalability (Bottou et al., 2018). Among them, *gradient descent* (GD), defined by

$$x_{k+1} = x_k - \alpha_k \nabla f(x_k), \tag{3}$$

is fundamental. Despite its simplicity, GD faces two main challenges:

- **Step size selection.** Convergence depends heavily on the step size $\alpha_k$. Small $\alpha_k$ slows progress; large $\alpha_k$ risks divergence. For $L$-smooth functions, $\alpha_k = 1/L$ is standard, but this global constant often mismatches local curvature.

- **Slow convergence.** Even with an optimal step size, GD is slow on ill-conditioned problems, i.e., when $L/\mu \gg 1$.

We briefly review strategies addressing these issues:

**Adaptive step sizes** Adaptive schemes such as the Barzilai–Borwein (BB) method (Barzilai and Borwein, 1988) estimate step sizes from past iterates:

$$\alpha_k = \frac{\langle x_k - x_{k-1}, \nabla f(x_k) - \nabla f(x_{k-1}) \rangle}{\|\nabla f(x_k) - \nabla f(x_{k-1})\|^2}, \tag{4}$$

with low computational overhead. However, BB-type methods are heuristic, and may diverge even for simple convex problems (Burdakov et al., 2019); guarantees are largely limited to quadratic cases (Dai and Liao, 2002). Extensions (Zhou et al., 2006; Dai et al., 2015) improve robustness but still lack general theory.

Polyak's method (Polyak, 1969), foundational to adaptive approaches such as AdaGrad and AMSGrad (Vaswani et al., 2020), ensures convergence but requires the optimal value $f^*$, which is rarely available.

**Acceleration** Momentum-based methods accelerate convergence by leveraging past updates. The heavy-ball method (Polyak, 1964) and Nesterov's accelerated gradient (NAG) (Nesterov, 2003) achieve the optimal rate $1 - \sqrt{\mu/L}$ under strong convexity, assuming known $L$ and $\mu$. In the convex case ($\mu = 0$), NAG with step size $1/(k+3)$ (Nesterov, 1983) achieves the optimal $O(1/k^2)$ rate. Nesterov later extended this framework to composite problems by incorporating line search into accelerated proximal methods (Nesterov, 2012), also attaining $O(1/k^2)$. Despite the effectiveness, NAG are known to suffer from oscillations. Restarting is used to mitigate this issue O'donoghue and Candes (2015).

**Backtracking line search** Backtracking line search begins with a large step size $\alpha_k$ and reduces it until conditions such as the Armijo–Goldstein criterion (Armijo, 1966; Goldstein, 1962/63) or Wolfe condition (Wolfe, 1969) are satisfied. Extensions (Ito and Fukuda, 2021; Liu and Yang, 2017) adapt line search to composite settings. Guminov et al. (2019) update parameters in NAG with backtracking, while Lan et al. (2023) develop a parameter-free method that attains optimal bounds for convex problems, and the best known results for nonconvex settings. An adaptive variant (Cavalcanti et al., 2025a) reduces backtracking steps, improving efficiency. Despite robustness and simplicity, line search usually requires 3-4 extra function or gradient evaluations per iteration, increasing cost.

**Line-search free methods.** Recent years have seen growing interest in line-search free adaptive methods. These algorithms keep the per-iteration cost of gradient descent while often achieving faster convergence and lower sensitivity to hyperparameters. Levy et al. (2018) incorporate AdaGrad-style adaptive step sizes into NAG and further extend to stochastic settings. However, their approach adapts only $L$, limiting its ability to go beyond standard NAG. Malitsky and Mishchenko (2020; 2024) introduced adaptive proximal gradient methods with theoretical guarantees, though lack of acceleration can hinder performance on ill-conditioned problems. Li and Lan (2024) and Cavalcanti et al. (2025b) proposed adaptive NAG variants with backtracking-free updates, in which both $L$ and $\mu$ are adaptive enabling stronger numerical performances.

In training deep neural networks, Adam (Adaptive Moment Estimation) (Kingma and Ba, 2015) is a widely used optimization method that combines momentum with adaptive step sizes to stabilize and accelerate stochastic gradients. However, the original Adam algorithm does not provide convergence guarantees, even in convex settings.

### CONTRIBUTION

- We develop A$^2$GD, an adaptive accelerated gradient method with provable accelerated linear convergence for smooth (1) and composite convex optimization (2).

- We adapt stability analysis from ODE solvers to reduce line search overhead, activating it only when accumulated perturbations are positive. The method is thus line-search reduced rather than line-search free (Fig. 2), and it outperforms existing line-search free methods in both theory and practice.

- We show numerically that A$^2$GD also consistently outperforms AGD variants (where a single A denotes either adaptivity or acceleration) and other methods combining adaptivity and acceleration.

**Limitations and Extensions** While A$^2$GD achieves adaptive acceleration with theoretical guarantees, these results rely on convexity, and extending the framework to nonconvex settings remains open. Empirically, the method still works once the iterate enters locally convex basins. We demonstrate the success of A$^2$GD on a composite $\ell_{1\text{-}2}$ problem, where the nonconvex regularizer admits a closed-form proximal operator.

Although our line-search reduced methods adds little practical overhead, we do not yet have a nontrivial upper bound on the total number of activations of line search as variability in local curvature for general convex functions can produce irregular triggering patterns. Empirically, typically fewer than 10, and almost all activations occur in the early phase.

Another extension is the stochastic setting. Developing a stochastic variant of A$^2$GD that preserves both adaptivity and acceleration under variance conditions would broaden

applicability to large-scale machine learning, providing a step toward a theoretical justification of the empirical success of Adam.

**Background on convex functions**  Let $f : \mathbb{R}^d \to \mathbb{R}$ be differentiable. The Bregman divergence between $x, y \in \mathbb{R}^d$ is defined as

$$D_f(y, x) := f(y) - f(x) - \langle \nabla f(x), y - x \rangle.$$

The function $f$ is $\mu$-strongly convex if for some $\mu > 0$,

$$D_f(y, x) \geq \frac{\mu}{2} \|y - x\|^2, \quad \forall x, y \in \mathbb{R}^d.$$

It is $L$-smooth, for some $L > 0$, if its gradient is $L$-Lipschitz:

$$\|\nabla f(y) - \nabla f(x)\| \leq L \|y - x\|, \quad \forall x, y \in \mathbb{R}^d.$$

The condition number is defined by $\kappa = L/\mu$. Let $\mathcal{S}_{L,\mu}$ denote the class of all differentiable functions that are both $\mu$-strongly convex and $L$-smooth.

For $f \in \mathcal{S}_{L,\mu}$, the Bregman divergence satisfies (Nesterov, 2003):

$$\frac{1}{2L} \|\nabla f(x) - \nabla f(y)\|^2 \leq D_f(x, y) \leq \frac{1}{2\mu} \|\nabla f(x) - \nabla f(y)\|^2, \quad \forall x, y \in \mathbb{R}^d. \tag{5}$$

Taking $y = x^*$, where $x^*$ minimizes $f$ and $\nabla f(x^*) = 0$, yields:

$$\|\nabla f(x)\|^2 \geq 2\mu(f(x) - f(x^*)), \quad \forall x \in \mathbb{R}^d. \tag{6}$$

## 2 ADAPTIVE GRADIENT DESCENT METHOD

We illustrate our main idea using gradient descent method (3) and later extend it to accelerated gradient descent. The steepest descent step chooses

$$\alpha_k^* = \arg \min_{\alpha > 0} f(x_k - \alpha \nabla f(x_k)), \tag{7}$$

which entails solving a one-dimensional convex problem. While conceptually simple, this can be costly unless a closed form is available.

For $L$-smooth functions, the fixed step size $\alpha_k = 1/L$ guarantees convergence, but is often overly conservative when local curvature is much smaller than $L$. To improve efficiency, we design step sizes that adapt to local geometry using $f(x_k)$ and $\nabla f(x_k)$.

We shall estimate the local Lipschitz constant $L_k$ through Lyapunov analysis of the gradient descent method (3). Consider the Lyapunov function

$$E_k = f(x_k) - f(x^\star), \tag{8}$$

where $x^\star \in \arg \min f(x)$ and $f(x^\star) = \min f$. Expanding $f$ at $x_{k+1}$ gives

$$\begin{aligned} E_{k+1} - E_k = f(x_{k+1}) - f(x_k) &= \langle \nabla f(x_{k+1}), x_{k+1} - x_k \rangle - D_f(x_k, x_{k+1}) \\ &= -\alpha_k \langle \nabla f(x_{k+1}), \nabla f(x_k) \rangle - D_f(x_k, x_{k+1}) \\ &= -\frac{\alpha_k}{2} \|\nabla f(x_{k+1})\|^2 - \frac{\alpha_k}{2} \|\nabla f(x_k)\|^2 \\ &\quad + \frac{\alpha_k}{2} \|\nabla f(x_{k+1}) - \nabla f(x_k)\|^2 - D_f(x_k, x_{k+1}). \end{aligned}$$

Applying (6) to $\|\nabla f(x_{k+1})\|^2$ and rearranging yields

$$(1 + \mu \alpha_k) E_{k+1} \leq E_k - \frac{\alpha_k}{2} \|\nabla f(x_k)\|^2 + \frac{\alpha_k}{2} \|\nabla f(x_{k+1}) - \nabla f(x_k)\|^2 - D_f(x_k, x_{k+1}). \tag{9}$$

If we use a line search to choose a small enough $\alpha_k$ such that

$$\alpha_k = \frac{1}{L_k} \leq \frac{2 D_f(x_k, x_{k+1})}{\|\nabla f(x_{k+1}) - \nabla f(x_k)\|^2}, \tag{10}$$

then dropping the negative terms in (9) gives the linear convergence

$$E_{k+1} \leq (1 + \mu/L_k)^{-1} E_k.$$

Since $\alpha_k = 1/L_k$, choosing a smaller $\alpha_k$ is equivalent to using a larger $L_k$. By (5), the criterion (10) holds once $L_k \geq L$. Standard backtracking starts with an initial estimate of $L_k$ and increases it iteratively by a factor $r > 1$ until (10) is satisfied. This procedure requires at most $\mathcal{O}(\lceil \log L / \log r \rceil)$ iterations. A more adaptive and efficient backtracking scheme was recently proposed in Cavalcanti et al. (2025a), which we adapt for our purposes and briefly recall below.

Rewriting the stopping criterion (10) gives

$$v = \frac{2 L_k D_f(x_k, x_{k+1})}{\|\nabla f(x_{k+1}) - \nabla f(x_k)\|^2} \geq 1.$$

If $v < 1$, the criterion is not satisfied. Instead of increasing $L_k$ by a fixed ratio, we update it as $L_k \leftarrow r L_k / v$, where $r > 1$ is a base ratio (e.g., $r = 3$). This adaptive scaling adjusts to the gap between the current condition and the stopping criterion, improving both efficiency and accuracy.

Even with adaptive backtracking, line search introduces overhead because each update of $L_k$ requires a new evaluation of $\nabla f(x_{k+1})$, often the dominant cost in gradient-based methods, and sometimes also $f(x_{k+1})$. To reduce this, recent work has increasingly focused on line-search–free adaptive schemes; see the literature review in the introduction.

Enforcing line-search free updates is often too rigid and restrictive. In contrast, we reduce the number of line-search steps, achieving comparable cost to line-search free methods. Our approach is inspired by stability analysis in ODE solvers. The following result can be easily established by induction.

**Lemma 2.1** (A variant of Lemma 5.7.1. in Gautschi (2011)). *Let $\{E_k\}$ be a positive sequence satisfying*

$$E_{k+1} \leq \delta_k(E_k + b_k), \quad k = 0, 1, \dots,$$

*where $\delta_k > 0$ and $b_k \in \mathbb{R}$. Then*

$$E_{k+1} \leq \left( \prod_{i=0}^{k} \delta_i \right) E_0 + p_k, \quad k = 0, 1, \dots,$$

*where the accumlated perturbation*

$$p_k = \sum_{i=0}^{k} \left( \prod_{j=i}^{k} \delta_j \right) b_i, \quad satisfying \quad p_k = \delta_k(p_{k-1} + b_k).$$

We use an adaptive gradient descent method (ad-GD) to illustrate our main idea and refer to Appendix A for the detailed algorithmic formulation. Applying Lemma 2.1 to GD under the Lyapunov analysis (9) gives

$$\delta_k = (1 + \mu/L_k)^{-1}, \quad b_k = b_k^{(1)} + b_k^{(2)}, \quad \text{where}$$

$$b_k^{(1)} = \frac{1}{2L_k} \|\nabla f(x_{k+1}) - \nabla f(x_k)\|^2 - D_f(x_k, x_{k+1}), \qquad b_k^{(2)} = -\frac{1}{2L_k} \|\nabla f(x_k)\|^2.$$

In the line-search criterion (10), $L_k$ is selected so that $b_k^{(1)} < 0$, ensuring $b_k < 0$ at each step. This pointwise condition is sufficient but not necessary. Instead, we activate line search only when $p_k > 0$ and increase $L_k$ until $p_k \leq 0$. Classical line search enforces $b_k < 0$ in an $\ell_\infty$ sense, while our approach permits a weighted $\ell_1$ control. Early in the iteration, when $\|\nabla f(x_k)\|$ is large, the negative terms $b_k^{(2)}$ accumulate and offset later positives, reducing activations. Once $p_k \leq 0$, exponential decay follows:

$$E_{k+1} \leq \prod_{i=0}^{k} \left( 1 + \frac{\mu}{L_i} \right)^{-1} E_0.$$

Figures 1 and 2 illustrate this behavior.

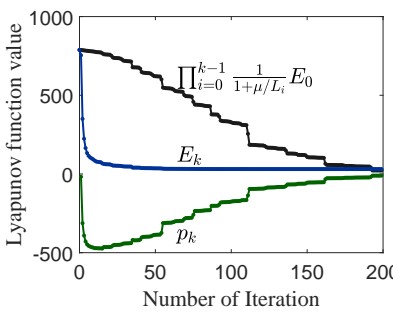

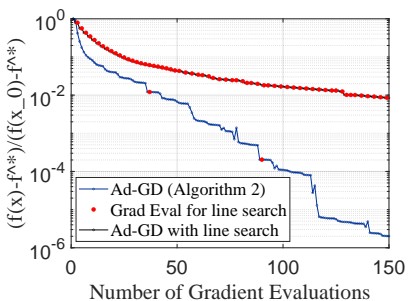

Figure 1: The accumulated perturbation $p_k$ (green) stays negative and approaches zero. The Lyapunov values $E_k$ (blue) decay faster than the theoretical exponential rate $\left(\prod_{i=0}^{k} \delta_i\right) E_0$ (black). In the early iterations, $E_k$ decreases even more rapidly due to the large negative term $b_k^{(2)} = -\frac{1}{2L_k}\|\nabla f(x_k)\|^2$.

Figure 2: For a logistic regression problem (14), gradient descent with line search enforcing $b_k^{(1)} \leq 0$ (top curve) triggers backtracking every 3–4 iterations on average. In contrast, ad-GD, which performs line search only when $p_k > 0$, requires far fewer activations (bottom curve). Red dots mark iterations where line search is triggered.

**Theorem 2.2.** *Assume $f \in \mathcal{S}_{L,\mu}$. Let $\{x_k\}$ be the sequence generated by gradient descent method (3) with line search ensuring $p_k \leq 0$. Then we have*

$$E_k \leq \prod_{i=0}^{k-1} \frac{1}{1 + \mu/L_i} E_0 \leq \left(\frac{1}{1 + \mu/(c_r L)}\right)^k E_0.$$

*Proof.* As $p_k \leq 0$ for all $k$, linear convergence follows from (9). By (5), the stopping criterion (10) is satisfied once $L_k \leq c_r L$ with at most $O(|\log L/\log r|)$ search steps, where $c_r \geq 1$ depends on the line-search scaling factor. Since $\mu_k \geq \mu$, the desired linear convergence rate follows. $\square$

**Remark 2.1.** To improve efficiency, we set the next step size as $\alpha_{k+1} = \frac{2D_f(x_k, x_{k+1})}{\|\nabla f(x_{k+1}) - \nabla f(x_k)\|^2}$. The gradient $\nabla f(x_{k+1})$ can be reused in the following gradient descent step. However, computing $D_f(x_k, x_{k+1})$ requires function evaluations $f(x_k)$ and $f(x_{k+1})$, which may be costly. To avoid these evaluations, we approximate $2D_f(x_k, x_{k+1})$ by its symmetrized form:

$$2D_f(x_k, x_{k+1}) \approx D_f(x_k, x_{k+1}) + D_f(x_{k+1}, x_k) = \langle \nabla f(x_{k+1}) - \nabla f(x_k),\ x_{k+1} - x_k \rangle.$$

This reduces the ratio to the form used in the BB gradient method (4). In contrast to BB, convergence of ad-GD is guaranteed by enforcing $p_k \leq 0$.

**Remark 2.2.** There are several variants depending on how we define $\delta_k$ and split $b_k^{(1)}$ and $b_k^{(2)}$. For example, we can use $\delta_k = 1 - \mu/L_k$, $b_k^{(2)} = 0$, and the rest is $b_k^{(1)}$. Then $b_k^{(1)} \leq 0$ is equivalent to the criteria proposed by (Nesterov, 2012) (Appendix A).

## 3    ADAPTIVE ACCELERATED GRADIENT DESCENT METHOD

In this section, we apply our adaptive strategy to accelerated gradient methods. We derive an identity for the difference of the Lyapunov function and adaptively adjust $L_k$ and $\mu_k$ to ensure the accumulated perturbation is non-positive.

We will use the Hessian-based Nesterov accelerated gradient (HNAG) flow proposed in Chen and Luo (2019)

$$\begin{cases} x' = y - x - \beta \nabla f(x), \\ y' = x - y - \dfrac{1}{\mu} \nabla f(x), \end{cases} \tag{11}$$

where $\beta$ is a positive parameter. An implicit and explicit (IMEX) discretization of (11) is

$$
\begin{cases}
x_{k+1} - x_k = \alpha_k \left( y_k - x_{k+1} \right) - \dfrac{1}{L_k} \nabla f(x_k), \\[2mm]
y_{k+1} - y_k = -\dfrac{\alpha_k}{\mu_k} \nabla f(x_{k+1}) + \alpha_k \left( x_{k+1} - y_{k+1} \right),
\end{cases}
\tag{12}
$$

where $\alpha_k > 0$ is the time step size and $L_k := (\alpha_k \beta_k)^{-1}$. Denote by $\boldsymbol{z} = (x, y)^{\mathsf{T}}$. Introduce the Lyapunov function

$$
\mathcal{E}(\boldsymbol{z}; \mu) := f(x) - f(x^*) + \frac{\mu}{2} \left\| y - x^* \right\|^2.
$$

The proof of the following identity can be found in Appendix B.

**Lemma 3.1.** *We have the identity*

$$
(1 + \alpha_k)\mathcal{E}(\boldsymbol{z}_{k+1}; \mu_k) - \mathcal{E}(\boldsymbol{z}_k; \mu_k)
$$

$$
= \frac{1}{2}\left( \frac{\alpha_k^2}{\mu_k} - \frac{1}{L_k} \right) \left\| \nabla f(x_{k+1}) \right\|_*^2 \quad \text{(I)}
$$

$$
+ \frac{1}{2L_k} \left\| \nabla f(x_{k+1}) - \nabla f(x_k) \right\|^2 - D_f(x_k, x_{k+1}) \quad \text{(II)}
$$

$$
- \frac{1}{2L_k} \left\| \nabla f(x_k) \right\|_*^2 + \frac{\alpha_k \mu_k}{2} \left( \left\| x_{k+1} - x^* \right\|^2 - \frac{1}{\mu_k} D_f(x^*, x_{k+1}) - (1 + \alpha_k) \left\| x_{k+1} - y_{k+1} \right\|^2 \right) \quad \text{(III)}.
$$

We can simply set $\alpha_k = \sqrt{\frac{\mu_k}{L_k}}$ so that (I) = 0. To control (II) and (III), define perturbations

$$
b_k^{(1)} = \frac{1}{2L_k} \left\| \nabla f(x_{k+1}) - \nabla f(x_k) \right\|^2 - D_f(x_k, x_{k+1}),
$$

$$
b_k^{(2)} = -\frac{1}{2L_k} \left\| \nabla f(x_k) \right\|^2 + \frac{\alpha_k \mu_k}{2} \left( R_k^2 - (1 + \alpha_k) \left\| x_{k+1} - y_{k+1} \right\|^2 \right),
\tag{13}
$$

$$
p_k = \frac{1}{1 + \alpha_k} \left( p_{k-1} + b_k^{(1)} + b_k^{(2)} \right), \quad \forall k \geq 1 \text{ and } p_0 = 0.
$$

The term $b_k^{(1)}$ measures deviation from the Lipschitz condition and is used to adjust $L_k$, while $b_k^{(2)}$ measures deviation from the strong convexity assumption and is used to adjust $\mu_k$. To enforce the lower bound $\mu_k \geq \mu$ when $\mu > 0$, we introduce

$$
R_k^2 := \left( 1 - \mu/\mu_k \right) R^2,
$$

using the inequality $D_f(x^*, x_{k+1}) \geq \frac{\mu}{2} \|x_{k+1} - x^*\|^2$ and an upper bound $R$ such that $\|x_{k+1} - x^*\|^2 \leq R^2$. If $\mu_k < \mu$, then $b_k^{(2)} < 0$ and no further reduction of $\mu_k$ is allowed. The parameter $\mu$ can be a conservative estimate of the true convexity constant and serves as a lower bound for $\mu_k$.

Line search is triggered only when $p_k > 0$. If $b_k^{(1)} > 0$, $L_k$ is updated using adaptive backtracking Cavalcanti et al. (2025a). If $b_k^{(2)} > 0$, the convexity is not strong enough to support a large step, so we reduce $\mu_k$. In the limiting case $\mu_k = 0$, we will have $b_k^{(2)} \leq 0$.

To update $\mu_k$ more precisely, we solve $b_k^{(2)} = 0$, treating $L_k$ as known and using the fixed rule $\alpha_k = \sqrt{\mu_k/L_k}$ for the step size. The leading term in the second part of $b_k^{(2)}$ is $\alpha_k \mu_k R_k^2 = \mu_k^{3/2} R_k^2 / L_k^{1/2}$, and the equation essentially reduces to a non-trivial scaling

$$
\frac{\mu_k^{3/2} R_k^2}{L_k^{1/2}} \approx \frac{\left\| \nabla f(x_k) \right\|^2}{L_k} \quad \Rightarrow \quad \mu_k \propto \frac{\left\| \nabla f(x_k) \right\|^{4/3}}{L_k^{1/3} R_k^{4/3}}.
$$

To preserve decay of the Lyapunov function, we enforce

$$
\mu_{k+1} \leq \mu_k \quad \Rightarrow \quad \mathcal{E}(\boldsymbol{z}_{k+1}; \mu_{k+1}) \leq \mathcal{E}(\boldsymbol{z}_{k+1}; \mu_k).
$$

To establish convergence guarantees, the parameter $\mu_k$ cannot decay too quickly. To control this decay, we follow the perturbation strategy of Chen et al. (2025) by introducing a parameter $\varepsilon$ and enforcing the lower bound $\mu_k \geq \varepsilon$. The value of $\varepsilon$ is halved only when certain decay conditions are met. Specifically, $\varepsilon$ is reduced if either $\mathcal{E}_k/\mathcal{E}_0 \leq (R^2 + 1)\varepsilon/2$ or the number of iterations performed with the current $\varepsilon$ exceeds $m$. If $\mu > 0$, the condition is reached within $\mathcal{O}(|\log \varepsilon|)$ iterations; if $\mu = 0$, the iteration count for a fixed $\varepsilon$ is at most $m$. Since $\mathcal{E}_k$ is not directly observable, we use the proxy $\|\nabla f(x_k)\|^2/\|\nabla f(x_0)\|^2$ for $\mathcal{E}_k/\mathcal{E}_0$.

---

**Algorithm 1:** Adaptive Accelerated Gradient Method ($\text{A}^2\text{GD}$)

---

**Input:** $x_0, y_0 \in \mathbb{R}^n$, $L_0 > 0$, $\mu_0 > 0$, $R > 0$, $0 < \text{tol} \ll 1$, $\varepsilon > 0$, $m \geq 1$

1  **while** $\|\nabla f(x_k)\| > \text{tol}\|\nabla f(x_0)\|$ **do**
2  $\quad \alpha_k \leftarrow \sqrt{\mu_k/L_k}$;
3  $\quad x_{k+1} \leftarrow \frac{1}{\alpha_k+1}x_k + \frac{\alpha_k}{\alpha_k+1}y_k - \frac{1}{L_k(\alpha_k+1)}\nabla f(x_k)$;
4  $\quad y_{k+1} \leftarrow \frac{\alpha_k}{\alpha_k+1}x_{k+1} + \frac{1}{\alpha_k+1}y_k - \frac{\alpha_k}{\mu_k(\alpha_k+1)}\nabla f(x_{k+1})$;
5  $\quad b_k^{(1)} \leftarrow \frac{1}{2L_k}\|\nabla f(x_{k+1}) - \nabla f(x_k)\|^2 - D_f(x_k, x_{k+1})$;
6  $\quad b_k^{(2)} \leftarrow -\frac{1}{2L_k}\|\nabla f(x_k)\|_*^2 + \frac{\alpha_k\mu_k}{2}\left(R_k^2 - (1+\alpha_k)\|x_{k+1} - y_{k+1}\|^2\right)$;
7  $\quad p_k \leftarrow \frac{1}{1+\alpha_k}(p_{k-1} + b_k^{(1)} + b_k^{(2)})$;
8  $\quad$ **if** $p_k > 0$ **then**
9  $\quad\quad$ **if** $b_k^{(1)} > 0$ **then**
10 $\quad\quad\quad v \leftarrow \frac{2L_k D_f(x_k, x_{k+1})}{\|\nabla f(x_{k+1})-\nabla f(x_k)\|^2}$, $L_k \leftarrow 3L_k/v$;
11 $\quad\quad$ **if** $b_k^{(2)} > 0$ **then**
12 $\quad\quad\quad \mu_k \leftarrow \max\left\{\varepsilon, \min\left\{\mu_k, \frac{\|\nabla f(x_k)\|^{4/3}}{L_k^{1/3}\left(R_k^2-(1+\alpha_k)\|x_{k+1}-y_{k+1}\|^2\right)^{2/3}}\right\}\right\}$;
13 $\quad\quad$ Go to line 2;
14 $\quad$ **else**
15 $\quad\quad L_{k+1} \leftarrow \frac{\|\nabla f(x_{k+1})-\nabla f(x_k)\|^2}{2D_f(x_k, x_{k+1})}$;
16 $\quad\quad \mu_{k+1} \leftarrow \max\left\{\varepsilon, \min\left\{\mu_k, \frac{\|\nabla f(x_k)\|^{4/3}}{L_k^{1/3}\left(R_k^2-(1+\alpha_k)\|x_{k+1}-y_{k+1}\|^2\right)^{2/3}}\right\}\right\}$;
17 $\quad$ **if** *decay condition* **then**
18 $\quad\quad \varepsilon \leftarrow \varepsilon/2$;
19 $\quad\quad m \leftarrow \lfloor\sqrt{2}\cdot m\rfloor + 1$;
20 $\quad k \leftarrow k + 1$;

---

To ensure monotonic descent, updates with $f(x_{k+1}) > f(x_k)$ are rejected by setting $x_{k+1} = x_k$. When $\|y_k - x^\star\| \gg \|x_k - x^\star\|$, the Lyapunov function $\mathcal{E}$ may decrease mainly through $\|y_k - x^\star\|^2$, while $f(x_k)$ stagnates. To avoid this, we restart by setting $y_k = x_k$ if $f(x_k)$ fails to decrease for five consecutive iterations. These monitoring steps are omitted from Algorithm 1 but are used in practice to improve stability. Restarting and accept/reject heuristics reduce oscillations but do not influence the main source of acceleration; see Fig. 3.

To reduce sensitivity to initialization, we include a short warm-up phase using adaptive proximal gradient descent (AdProxGD) Malitsky and Mishchenko (2024) or ad-GD (Algorithm 2 in Appendix A). Starting from $x_0$, we run 10 AdProxGD iterations and initialize $\text{A}^2\text{GD}$ with $x_0 = y_0 := x_{10}$, $\mu_0 := \min_{1\leq k\leq 10}\{L_k\}$, and $R = 100\|\nabla f(x_0)\|/\mu_0$. Although updating $R$ dynamically may help, the method is generally robust with a fixed $R$. Our ablation study shows that $\text{A}^2\text{GD}$ remains stable for perturbation up to a factor 1000, indicating that the warm-up is not essential for good performance; see Fig. 10.

**Theorem 3.2.** *Let $(x_k, y_k)$ be the iterates generated by the above algorithm. Assume function $f$ is $\mu$-strongly convex with $\mu \geq 0$. Let $k_s$ be the total number of steps after halving $\varepsilon$ exactly $s$ times, i.e. $\varepsilon = 2^{-s}\varepsilon_0$.*

1. *When $\mu = 0$, ther exists a constant $C > 0$ so that*

$$\frac{\mathcal{E}_{k_s}}{\mathcal{E}_0} \leq \frac{R^2 + 1}{\left(Ck_s + \varepsilon_0^{-1/2}\right)^2} = \mathcal{O}\left(\frac{1}{k_s^2}\right)$$

*So $\mathcal{O}(\sqrt{1/\mathrm{tol}})$ iteration steps can acheive $\mathcal{E}_{k_s}/\mathcal{E}_0 \leq \mathrm{tol}$.*

2. *When $\mu > 0$, the iteration number to achieve $\mathcal{E}_{k_s}/\mathcal{E}_0 \leq (R^2 + 1)2^{-s}\varepsilon_0 \leq \mathrm{tol}$ is bounded by $\mathcal{O}(\sqrt{L/\mu} \, \ln \mathrm{tol})$.*

## 4 NUMERICAL EXPERIMENTS

We test $A^2GD$ on smooth convex minimization tasks and compare it with several leading first-order methods, grouped into two categories:

- **Accelerated but non-adaptive methods:** Nesterov's accelerated gradient (NAG) with step size $1/(k+3)$ (Nesterov, 1983), accelerated over-relaxation heavy ball (AOR-HB) (Wei and Chen, 2025), and the triple momentum method (TM) (Van Scoy et al., 2018).
- **Adaptive methods:** adaptive proximal gradient descent (AdProxGD) (Malitsky and Mishchenko, 2024), the Accelerated Adaptive Gradient Method (AcceleGrad) (Levy et al., 2018), and NAGfree (Cavalcanti et al., 2025a).

For all examples, we set the tolerance to $\mathrm{tol} = 10^{-6}$ and use the stopping criterion $\|\nabla f(x_k)\| \leq \mathrm{tol} \cdot \|\nabla f(x_0)\|$. All experiments were run in MATLAB R2023a on a desktop with an Intel Core i5-7200U CPU (2.50 GHz) and 8 GB RAM. Because gradient evaluation dominates the computational cost, we report convergence in terms of gradient evaluations and mark additional evaluations from line search with red dots. The corresponding runtime comparisons are provided in Appendix D and the performance is similar.

**Regularized Logistic Regression** We report numerical simulations on a logistic regression problem with an $\ell_2$ regularizer:

$$\min_{x \in \mathbb{R}^n} \left\{ \sum_{i=1}^m \log\left(1 + \exp(-b_i a_i^\top x)\right) + \frac{\lambda}{2}\|x\|^2 \right\}, \tag{14}$$

where $(a_i, b_i) \in \mathbb{R}^n \times \{-1, 1\}$ for $i = 1, 2, \ldots, m$.

For this problem, $\mu = \lambda$ and $L = \lambda_{\max}\left(\sum_{i=1}^m a_i a_i^\top\right) + \lambda$. We use $(a_i, b_i)$ from the Adult Census Income dataset. After removing entries with missing values, the dataset contains 30,162 samples. The Lipschitz constant is $6.30 \times 10^4$. With regularization parameter $\lambda = 0.1$, the condition number is $\kappa = 6.30 \times 10^5$.

In Fig. 3, we disable accept/reject and restarting in $A^2GD$ and obtain $A^2GD$-plain. We compare it with other plain accelerated methods. In Fig. 4, we equip all methods with comparable restarting schemes and evaluate them alongside $A^2GD$. In Fig. 5, we compare $A^2GD$ with other fine-tuned adaptive gradient methods using their recommended parameter settings.

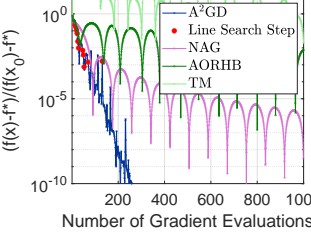

Figure 3: Comparison without restarting.

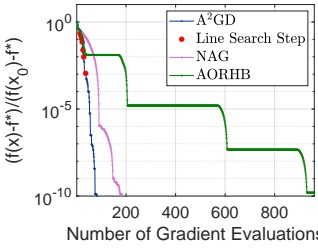

Figure 4: Comparison with restarting.

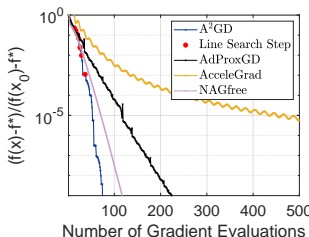

Figure 5: Comparison with other adaptive methods.

**Maximum Likelihood Estimate of the Information Matrix**  We consider the maximum likelihood estimation problem from (Boyd and Vandenberghe, 2004, (7.5)):

$$\underset{X \in \mathbb{R}^{n \times n}}{\text{minimize}} \quad f(X) := -\log \det X + \text{tr}(XY),$$
$$\text{subject to} \quad \lambda_{\min} \leq \lambda(X) \leq \lambda_{\max}, \tag{15}$$

where $X$ is symmetric positive definite and $\lambda_{\min}, \lambda_{\max} > 0$ are given bounds. The condition number of $f$ is $\kappa = \lambda_{\max}^2 / \lambda_{\min}^2$.

Problem (15) has a composite structure, with a smooth term $f(X)$ and a nonsmooth indicator $g(X)$ enforcing spectral constraints. The proximal step for $g$ requires eigen-decomposition, eigenvalue projection, and matrix reconstruction, so gradient and proximal evaluations dominate the cost. Reducing these evaluations, particularly during backtracking, is therefore crucial. We again report convergence in terms of gradient steps and mark additional line-search evaluations with red dots, which occur very rarely after the initial stage and thus invisable in the figures.

We extend A$^2$GD and its convergence analysis to the composite setting; details appear in Appendix C. We compare A$^2$GD with several first-order proximal methods: AdProxGD (Malitsky and Mishchenko, 2024), FISTA (Beck and Teboulle, 2009), and AOR-HB with perturbation (Chen et al., 2025). We use tolerance $\text{tol} = 10^{-6}$ and adopt the stopping rule $\|\nabla h(x_k) + q_k\| \leq \text{tol} \|\nabla h(x_0)\|$ for all experiments.

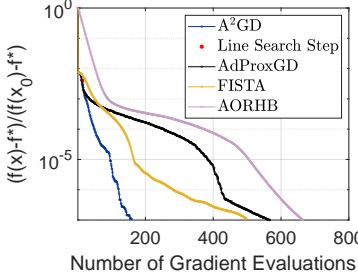

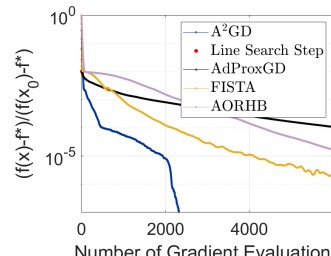

Figure 6: Error curves under setting (1).

Figure 7: Error curves under setting (2).

Following Malitsky and Mishchenko (2024), we construct the data matrix $Y$ as follows: sample a random vector $y \in \mathbb{R}^n$, and define $y_i = y + \delta_i$ for $i = 1, \ldots, M$, with $\delta_i \sim \mathcal{N}(0, I_n)$. Then set $Y = \frac{1}{M} \sum_{i=1}^{M} y_i y_i^\top$. We test our algorithm under two settings: (1) $n = 100$, $M = 50$, $\lambda_{\min} = 0.1$, $\lambda_{\max} = 10$; (2) $n = 50$, $M = 100$, $\lambda_{\min} = 0.1$, $\lambda_{\max} = 10^3$.

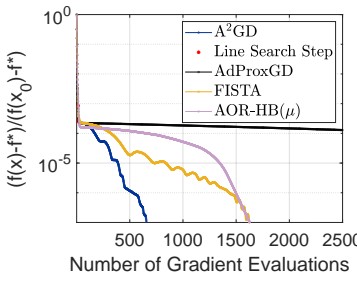

Figure 8: Error curve for $\ell_{1\text{-}2}$ problem with $n = 500, p = 1000$.

**$\ell_{1\text{-}2}$ nonconvex minimization problem**  We consider the $\ell_{1\text{-}2}$ minimization problem

$$\min_{x \in \mathbb{R}^n} \quad \frac{1}{2} \|Ax - b\|^2 + \lambda(\|x\|_1 - \|x\|_2), \tag{16}$$

introduced by Yin et al. (2015), promotes sparser solutions than standard convex penalties.

The matrix $A \in \mathbb{R}^{n \times p}$ is generated from a standard Gaussian distribution, and the ground truth $x^* \in \mathbb{R}^p$ has sparsity 50. The observation vector is constructed as $b = Ax^*$. We set the regularization parameter $\lambda = 1$ and the problem size: $n = 500$, $p = 1000$. The initial point is sampled as $x_0 = y_0 \sim 10\mathcal{N}(0, I_p)$.

**Scaling behavior.**  We consider the linear finite element method for the Poisson problem

$$-\Delta u = b \text{ in } \Omega, \qquad u = 0 \text{ on } \partial\Omega,$$

where $\Omega$ is the unit disk discretized by a quasi-uniform triangulation $\mathcal{T}_h$.

Table 1: Performance comparison on 2D linear Laplacian problem, tol $= 10^{-6}$

| Problem Size | | | A²GD | | AdProxGD | | NAG | | AOR-HB | |
|---|---|---|---|---|---|---|---|---|---|---|
| $h$ | $n$ | $\kappa$ | #Grad | Time | #Grad | Time | #Grad | Time | #Grad | Time |
| 1/20 | 1262 | 7.85e+02 | 162 | 0.02 | 895 | 0.05 | 583 | 0.02 | 248 | 0.01 |
| 1/40 | 5166 | 3.15e+03 | 293 | 0.10 | 3191 | 0.90 | 1205 | 0.26 | 418 | 0.11 |
| 1/80 | 20908 | 1.30e+04 | 476 | 0.65 | 10729 | 15.72 | 1651 | 1.83 | 699 | 0.81 |
| 1/160 | 84120 | 5.32e+04 | 791 | 5.06 | (>20000) | 131.19 | 2902 | 14.51 | 1187 | 6.11 |

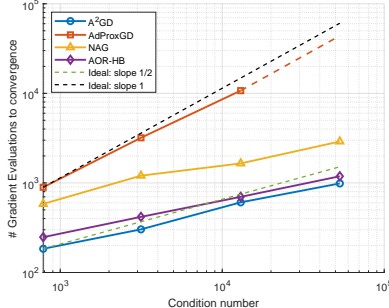

Figure 9: A²GD, NAG, and AOR-HB exhibit the accelerated $\sqrt{\kappa}$ scaling, whereas AdProxGD follows the non-accelerated $\kappa$ rate.

Using the *iFEM* package Chen (2009), we assemble the stiffness matrix $A$ and define the quadratic objective

$$f(x) = \tfrac{1}{2}(x - x^*)^\top A(x - x^*),$$

with $x^* \in \mathbb{R}^n$ and $x_0 \sim \text{Unif}(0, 1)$ componentwise.

It is well known that $\kappa(A) = O(h^{-2}) = O(n)$. We estimate $L$ and $\mu$ using the extreme eigenvalues of $A$. We use quasi-uniform meshes on the disk rather than structured square grids, where closed-form eigenvalue bounds are available and adaptive selection of $L$ and $\mu$ is less critical.

Table 1 shows that when the condition number increases by a factor of 4, the number of gradient steps for accelerated methods grows by roughly a factor of 2, and the total runtime by about a factor of 8, since each gradient evaluation becomes 4 times more expensive.

Figure 4 compares scaling behavior across methods: accelerated methods exhibit slopes near 1/2, whereas the non-accelerated AdProxGD scales with slope close to 1.

**Ablation study on the warm-up phase and hyper-parameters.** We include an adaptive gradient-descent warm-up in A²GD to automatically select $\mu_0$ and $R$, although this is not essential, since the method degrades only moderately when these hyper-parameters are misspecified. In experiments on the regularized logistic regression problem, A²GD remains robust even when these parameters vary by factors of $10^3$; see Fig. 10.

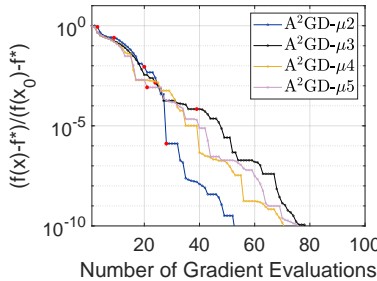
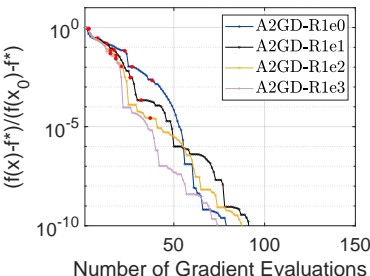

Figure 10: Comparison of A²GD variants without warm-up using manual initializations. The left panel shows the effect of different choices of $\mu_0$, and the right panel shows the effect of varying $R$. For $2 \leq i \leq 5$, "A²GD-$\mu_i$" denotes the manual choice $\mu_0 = 10^{-i}$, with $i = 2$ closest to the warm-up value; for $0 \leq j \leq 3$, "A²GD-R1e$j$" denotes the manual choice $R = 10^j$, with $j = 0$ matching the warm-up.

Across all tests, our A²GD method consistently outperforms baseline algorithms. Empirically, line search is triggered only a few times, typically fewer than 10, and almost all activations occur in the early phase. In the middle and late stages, line search are rare or completely absent across all tested problems.

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

## Appendix A: Adaptive Gradient Descent Method

We present an algorithm for adaptive gradient descent method (Ad-GD) which is a simplified version of $A^2$GD without momentum.

---

**Algorithm 2:** Adaptive Gradient Descent Method (Ad-GD)

---

**Input:** Initial point $x_0 \in \mathbb{R}^n$, initial step size $L_0 > 0$, initial strong convexity estimate $\mu_0 > 0$

**Output:** Sequence $\{x_k\}$

1 **for** $k = 0, 1, 2, \ldots$ **do**

2    $x_{k+1} \leftarrow x_k - \frac{1}{L_k}\nabla f(x_k)$;

3    $b_k^{(1)} \leftarrow \frac{1}{2L_k}\|\nabla f(x_{k+1}) - \nabla f(x_k)\|^2 - D_f(x_k, x_{k+1})$;

4    $b_k^{(2)} \leftarrow -\frac{1}{2L_k}\|\nabla f(x_k)\|^2$;

5    $p_k \leftarrow \left(1 + \frac{\mu_k}{L_k}\right)^{-1}\left(p_{k-1} + b_k^{(1)} + b_k^{(2)}\right)$;

6    **if** $p_k > 0$ **then**

7      Use adaptive backtracking to update $L_k$ so that $b_k^{(1)} \leq 0$;

8    $L_k \leftarrow \dfrac{\|\nabla f(x_k) - \nabla f(x_{k+1})\|^2}{2D_f(x_k, x_{k+1})}$;

9    $\mu_k \leftarrow \min\{\mu_k, L_k\}$;

---

There are several variants of Ad-GD depending on how we define $\delta_k$ and split $b_k^{(1)}$ and $b_k^{(2)}$. For example, we can use $\delta_k = 1 - \mu/L_k$, $b_k^{(2)} = 0$, and

$$b_k^{(1)} := \frac{1}{2L_k}\|\nabla f(x_{k+1}) - \nabla f(x_k)\|^2 - D_f(x_k, x_{k+1}) - \frac{1}{2L_k}\|\nabla f(x_{k+1})\|^2. \qquad (17)$$

The inequality $b_k^{(1)} \leq 0$ is equivalent to the criteria proposed by Nesterov in Nesterov (2012).

**Proposition 4.1.** *The inequality $b_k^{(1)} \leq 0$ is equivalent to*

$$m_{L_k}(x_{k+1}; x_k) \geq f(x_{k+1}), \qquad (18)$$

*where $m_{L_k}(y; x) = f(x) + \langle\nabla f(x), y - x\rangle + \frac{L_k}{2}\|y - x\|^2$, $x_{k+1} = \arg\min_y m_{L_k}(y; x_k)$.*

*Proof.* First, we have the identity

$$f(x_k) - f(x_{k+1}) - \frac{\alpha_k}{2}\|\nabla f(x_k)\|^2$$
$$= \frac{\alpha_k}{2}\|\nabla f(x_{k+1})\|^2 - \frac{\alpha_k}{2}\|\nabla f(x_{k+1}) - \nabla f(x_k)\|^2 + D_f(x_k, x_{k+1}).$$

Notice that $x_{k+1} = \arg\min_y m_{L_k}(y; x_k) \Leftrightarrow x_{k+1} = x_k - \frac{1}{L_k}\nabla f(x_k)$, so

$$m_{L_k}(x_{k+1}; x_k) \geq f(x_{k+1})$$

$$\Leftrightarrow f(x_k) + \left\langle \nabla f(x_k), -\frac{1}{L_k}\nabla f(x_k) \right\rangle + \frac{L_k}{2}\left\| \frac{1}{L}\nabla f(x_k) \right\|^2 \geq f(x_{k+1})$$

$$\Leftrightarrow f(x_k) - f(x_{k+1}) - \frac{1}{2L_k}\|\nabla f(x_k)\|^2 \geq 0$$

$$\Leftrightarrow \frac{1}{2L_k}\|\nabla f(x_{k+1})\|^2 + \frac{1}{2L_k}\|\nabla f(x_{k+1}) - \nabla f(x_k)\|^2 - D_f(x_k, x_{k+1}) \geq 0.$$

Thus, equivalence is proved. $\qquad\square$

## APPENDIX B: IDENTITIES OF ACCELERATED GRADIENT METHODS

We will use the Hessian-based Nesterov accelerated gradient (HNAG) flow proposed in Chen and Luo (2019)

$$\begin{cases} x' = y - x - \beta\nabla f(x), \\ y' = x - y - \frac{1}{\mu}\nabla f(x). \end{cases} \tag{19}$$

Denote by $\boldsymbol{z} = (x, y)^\intercal$ and $\mathcal{G}(\boldsymbol{z})$ the right hand side of (11), which now becomes $\boldsymbol{z}' = \mathcal{G}(\boldsymbol{z})$. In the notation $\nabla\mathcal{E}$, we consider $\mu$ as a fixed parameter and take derivative with respect to $\boldsymbol{z}$.

**Lemma 4.2.** *We have the identity*

$$-\nabla\mathcal{E}(\boldsymbol{z}) \cdot \mathcal{G}(\boldsymbol{z}) = \mathcal{E}(\boldsymbol{z}) + \beta\|\nabla f(x)\|_*^2 + \frac{\mu}{2}\|y - x\|^2 + D_f(x^*, x) - \frac{\mu}{2}\|x - x^\star\|^2. \tag{20}$$

*Proof.* A direct computation gives

$$-\nabla\mathcal{E}(\boldsymbol{z}) \cdot \mathcal{G}(\boldsymbol{z}) = \begin{pmatrix} \nabla f(x) \\ \mu(y - x^\star) \end{pmatrix}\begin{pmatrix} (x - x^\star) - (y - x^\star) + \beta\nabla f(x) \\ (y - x^\star) - (x - x^\star) + \frac{1}{\mu}\nabla f(x) \end{pmatrix}$$

$$= \langle\nabla f(x), x - x^\star\rangle + \beta\|\nabla f(x)\|_*^2 + \mu\|y - x^\star\|^2 - \mu(y - x^\star, x - x^\star) \tag{21}$$

$$= \mathcal{E}(\boldsymbol{z}) + \beta\|\nabla f(x)\|_*^2 + D_f(x^*, x) + \frac{\mu}{2}\|y - x\|^2 - \frac{\mu}{2}\|x - x^\star\|^2.$$

$\qquad\square$

**Lemma 4.3.** *We have the identity*

$$(1 + \alpha_k)\mathcal{E}(\boldsymbol{z}_{k+1}; \mu_k) - \mathcal{E}(\boldsymbol{z}_k; \mu_k)$$

$$= \text{(I)} \quad \frac{1}{2}\left(\frac{\alpha_k^2}{\mu_k} - \frac{1}{L_k}\right)\|\nabla f(x_{k+1})\|_*^2$$

$$\text{(II)} + \frac{1}{2L_k}\|\nabla f(x_{k+1}) - \nabla f(x_k)\|^2 - D_f(x_k, x_{k+1})$$

$$\text{(III)} - \frac{1}{2L_k}\|\nabla f(x_k)\|_*^2 + \frac{\alpha_k\mu_k}{2}\left(\|x_{k+1} - x^\star\|^2 - \frac{2}{\mu_k}D_f(x^\star, x_{k+1}) - (1 + \alpha_k)\|x_{k+1} - y_{k+1}\|^2\right).$$

*Proof.* Treat $\mu_k$ as a fixed parameter. We expand the difference

$$\mathcal{E}(\boldsymbol{z}_{k+1}; \mu_k) - \mathcal{E}(\boldsymbol{z}_k; \mu_k) = \langle\nabla\mathcal{E}(\boldsymbol{z}_{k+1}; \mu_k), \boldsymbol{z}_{k+1} - \boldsymbol{z}_k\rangle - D_\mathcal{E}(\boldsymbol{z}_k, \boldsymbol{z}_{k+1}; \mu_k), \tag{22}$$

where the negative term $-D_\mathcal{E}(\boldsymbol{z}_k, \boldsymbol{z}_{k+1}; \mu_k)$ is expanded as $-D_f(x_k, x_{k+1}) - \frac{\mu_k}{2}\|y_k - y_{k+1}\|^2$. Using the identity (21) in the continuous level, we have

$$\langle\nabla\mathcal{E}(\boldsymbol{z}_{k+1}; \mu_k), \alpha_k\mathcal{G}(\boldsymbol{z}_{k+1}, \mu_k)\rangle = -\alpha_k\mathcal{E}(\boldsymbol{z}_{k+1}, \mu_k)$$

$$- \frac{1}{L_k}\|\nabla f(x_{k+1})\|_*^2 - \alpha_k D_f(x^*, x_{k+1}) + \frac{\alpha_k\mu_k}{2}\left(\|x_{k+1} - x^\star\|^2 - \|x_{k+1} - y_{k+1}\|^2\right).$$

The difference between the scheme and the implicit Euler method is

$$\boldsymbol{z}_{k+1} - \boldsymbol{z}_k - \alpha_k \mathcal{G}(\boldsymbol{z}_{k+1}, \mu_k) = \alpha_k \begin{pmatrix} y_k - y_{k+1} + \beta_k(\nabla f(x_{k+1}) - \nabla f(x_k)) \\ 0 \end{pmatrix}.$$

which will bring more terms

$$\langle \nabla_x \mathcal{E}(\boldsymbol{z}_{k+1}, \mu_k), \boldsymbol{z}_{k+1} - \boldsymbol{z}_k - \alpha_k \mathcal{G}(\boldsymbol{z}_{k+1}, \mu_k) \rangle$$
$$= \frac{1}{L_k} \langle \nabla f(x_{k+1}), \nabla f(x_{k+1}) - \nabla f(x_k) \rangle + \alpha_k \langle \nabla f(x_{k+1}), y_k - y_{k+1} \rangle.$$

We then use the identity of squares for the cross term of gradients

$$\frac{1}{L_k} \langle \nabla f(x_{k+1}), \nabla f(x_{k+1}) - \nabla f(x_k) \rangle$$
$$= -\frac{1}{2L_k} \|\nabla f(x_k)\|_*^2 + \frac{1}{2L_k} \|\nabla f(x_{k+1})\|_*^2 + \frac{1}{2L_k} \|\nabla f(x_{k+1}) - \nabla f(x_k)\|_*^2.$$

As expected, this cross term brings more positive squares but also contribute a negative one.

On the second term, we write as

$$\alpha_k \langle \nabla f(x_{k+1}), y_k - y_{k+1} \rangle = \left\langle \frac{\alpha_k}{\sqrt{\mu_k}} \nabla f(x_{k+1}), \sqrt{\mu_k}(y_k - y_{k+1}) \right\rangle$$
$$= \frac{\alpha_k^2}{2\mu_k} \|\nabla f(x_{k+1})\|_*^2 + \frac{\mu_k}{2} \|y_k - y_{k+1}\|^2 - \frac{1}{2} \left\| \frac{\alpha_k}{\sqrt{\mu_k}} \nabla f(x_{k+1}) - \sqrt{\mu_k}(y_k - y_{k+1}) \right\|^2$$
$$= \frac{\alpha_k^2}{2\mu_k} \|\nabla f(x_{k+1})\|_*^2 + \frac{\mu_k}{2} \|y_k - y_{k+1}\|^2 - \frac{1}{2} \alpha_k^2 \mu_k \|x_{k+1} - y_{k+1}\|^2.$$

Combining altogether, we get the desired identity. $\qquad \square$

PROOF OF THEOREM 3.2

First, we prove convergence of Algorithm 1 within a single inner iteration, i.e. $\varepsilon$ is fixed, in the following lemma. It bears similarity to (Chen et al., 2025, Theorem 8.3), and is a direct result of Lemma 4.3.

**Lemma 4.4.** *Suppose $f$ is convex and $L$-smooth. Let $z_k = (x_k, y_k)$ be the iterates generated by Algorithm 1 within an inner iteration where $\mu = \varepsilon$. Assume that there exists $R > 0$ such that*

$$\|x_k - x^*\| \leq R, \quad \forall k \geq 0,$$

*and that there exists $l \in (\varepsilon, L)$ such that $L_k \geq l$ for all $k \geq 0$. Then the Lyapunov function exhibits linear convergence up to a perturbation:*

$$\mathcal{E}(z_k; \varepsilon) \leq \left( \frac{1}{1 + \sqrt{\varepsilon/(rL)}} \right)^k \mathcal{E}(z_0; \varepsilon) + \frac{\varepsilon}{2} R^2,$$

*where $r$ is the backtracking ratio (in Algorithm 1, $r = 3$).*

*Proof.* By Lemma 4.3, we have

$$\mathcal{E}(z_{k+1}; \mu_{k+1}) \leq \frac{1}{1 + \alpha_k} \mathcal{E}(z_k; \mu_k) + \frac{1}{1 + \alpha_k} \left( b_k^{(1)} + b_k^{(2)} \right) + \frac{\alpha_k \mu_k}{2(1 + \alpha_k)} R^2.$$

Since $l \leq L_k \leq rL$, it follows that

$$\sqrt{\frac{\varepsilon}{rL}} \leq \alpha_k \leq \sqrt{\frac{\varepsilon}{l}}.$$

Therefore,

$$\mathcal{E}(z_{k+1}; \mu_{k+1}) \leq \frac{1}{1 + \sqrt{\varepsilon/(rL)}} \mathcal{E}(z_k; \mu_k) + \frac{1}{1 + \alpha_k} \left( b_k^{(1)} + b_k^{(2)} \right) + \frac{\varepsilon \sqrt{\varepsilon/l}}{2(1 + \sqrt{\varepsilon/l})} R^2.$$

Iterating the inequality yields

$$\mathcal{E}(z_{k+1}) \leq \left(\frac{1}{1+\sqrt{\varepsilon/(rL)}}\right)^{k+1} \mathcal{E}(z_0) + p_{k+1} + \frac{\varepsilon\sqrt{\varepsilon/l}}{2\left(1+\sqrt{\varepsilon/l}\right)} \sum_{i=0}^{k} \left(\frac{1}{1+\sqrt{\varepsilon/l}}\right)^i R^2,$$

where $p_{k+1}$ is the accumulated perturbation. By Algorithm 1, we have $p_{k+1} \leq 0$.

Finally, the geometric sum is bounded as

$$\sum_{i=0}^{k} \left(\frac{1}{1+\sqrt{\varepsilon/l}}\right)^i \leq \frac{1+\sqrt{\varepsilon/l}}{\sqrt{\varepsilon/l}}.$$

Substituting this estimate gives the claimed bound

$$\mathcal{E}(z_{k+1};\varepsilon) \leq \left(\frac{1}{1+\sqrt{\varepsilon/(rL)}}\right)^{k+1} \mathcal{E}(z_0;\varepsilon) + \frac{\varepsilon}{2}R^2.$$

$\square$

*Proof of Theorem 3.2.* We distinguish between the convex case ($\mu = 0$) and the strongly convex case ($\mu > 0$).

If $\mu = 0$, in this case, the proof of (Chen et al., 2025, Theorem 8.4) applies directly, once the single-inner-iteration convergence relation (Lemma 4.4) is established. Therefore, no further argument is needed.

If instead, $\mu > 0$, recall that in the algorithm the effective radius is updated as

$$R_k^2 = \left(1 - \frac{\mu}{\mu_k}\right)R^2.$$

Thus, whenever $\mu_k \geq \mu$, we obtain $R_k^2 \leq 0$, which implies that further reduction of $\mu_k$ is no longer admissible. In particular, $\mu_k$ will stop decreasing once the tolerance parameter $\varepsilon$ satisfies $\varepsilon \leq \mu$.

Since $\varepsilon$ is halved at each outer stage, the final value of $\mu_k$ is therefore bounded below by $\mu/2$. At the same time, the smoothness parameter satisfies $L_k \leq rL$ by construction. Hence, in the terminal stage we obtain an effective condition number bounded by

$$\kappa_{\text{eff}} = \frac{L_k}{\mu_k} \leq \frac{rL}{\mu/2} = \frac{2rL}{\mu}.$$

Applying the convergence estimate from Lemma 4.4 in this regime, the Lyapunov function contracts linearly:

$$\mathcal{E}_{k_s} \leq \left(\frac{1}{1+\sqrt{\mu_k/L_k}}\right)^{k_s} \mathcal{E}_0 \leq \left(\frac{1}{1+\sqrt{\mu/2rL}}\right)^{k_s} \mathcal{E}_0.$$

Therefore, to ensure that $\mathcal{E}_{k_s} \leq \text{tol} \cdot \mathcal{E}_0$, it suffices to take

$$k_s \geq \frac{\ln(1/\text{tol})}{\ln\left(1+\sqrt{\mu/2rL}\right)} = \mathcal{O}\left(\sqrt{2rL/\mu}\,\ln(1/\text{tol})\right).$$

This establishes the desired complexity bound in both cases. $\square$

## Appendix C: Composite Convex Optimization

We derive the continuous time analogy to Lemma 3.1. First, define the composite right hand side update

$$\mathcal{G}(z) = \left(y - x - \beta(\nabla h(x) + q),\, x - y - \frac{1}{\mu}(\nabla h(x) + q)\right)^{\text{T}},$$

where $q \in \partial g(x)$. Let $\mathcal{E}_h(z;\mu) = h(x) - h(x^*) + \frac{\mu}{2}\|y - x^*\|^2$, then $\mathcal{E}(z;\mu) = \mathcal{E}_h(z;\mu) + (g(x) - g(x^*))$ is splitted into a smooth part and a non-smooth part.

**Lemma 4.5.** *We have the following inequality*

$$-\left\langle \nabla \mathcal{E}_h(x) + \begin{pmatrix} q \\ 0 \end{pmatrix}, \mathcal{G}(z) \right\rangle \geq \mathcal{E}(z) + \beta \|\nabla h(x) + q\|_*^2 + \frac{\mu}{2}\|y - x\|^2 + D_h(x^*, x) - \frac{\mu}{2}\|x - x^*\|^2.$$

*Proof.* A direct computation gives

$$-\left\langle \nabla \mathcal{E}_h(x) + \begin{pmatrix} q \\ 0 \end{pmatrix}, \mathcal{G}(z) \right\rangle = \begin{pmatrix} \nabla h(x) + q \\ \mu(y - x^\star) \end{pmatrix} \begin{pmatrix} (x - x^\star) - (y - x^\star) + \beta(\nabla h(x) + q) \\ (y - x^\star) - (x - x^\star) + \frac{1}{\mu}(\nabla h(x) + q) \end{pmatrix}$$

$$= \langle \nabla h(x) + q, x - x^\star \rangle + \beta \|\nabla h(x) + q\|_*^2 + \mu \|y - x^\star\|^2 - \mu(y - x^\star, x - x^\star) \qquad (23)$$

$$\geq \mathcal{E}(\boldsymbol{z}) + \beta \|\nabla h(x) + q\|_*^2 + D_h(x^*, x) + \frac{\mu}{2}\|y - x\|^2 - \frac{\mu}{2}\|x - x^\star\|^2,$$

the last inequality following from $q \in \partial g(x)$. $\qquad \square$

**Lemma 4.6.** *We have the following inequality*

$$(1 + \alpha_k)\mathcal{E}(\boldsymbol{z}_{k+1}; \mu_k) - \mathcal{E}(\boldsymbol{z}_k; \mu_k)$$

$$\leq \text{(I)} \quad \frac{1}{2}\left(\frac{\alpha_k^2}{\mu_k} - \frac{1}{L_k}\right)\|\nabla h(x_{k+1}) + q_{k+1}\|_*^2$$

$$\text{(II)} + \frac{1}{2L_k}\|\nabla h(x_{k+1}) - \nabla h(x_k)\|^2 - D_h(x_k, x_{k+1})$$

$$\text{(III)} - \frac{1}{2L_k}\|\nabla h(x_k) + q_{k+1}\|_*^2 + \frac{\alpha_k \mu_k}{2}\left(\|x_{k+1} - x^\star\|^2 - \frac{2}{\mu_k}D_h(x^\star, x_{k+1}) - (1 + \alpha_k)\|x_{k+1} - y_{k+1}\|^2\right).$$

*Proof.* The proof is similar to the smooth convex case. Expand the difference of $\mathcal{E}$ at $z_{k+1}$,

$$\mathcal{E}(\boldsymbol{z}_{k+1}; \mu_k) - \mathcal{E}(\boldsymbol{z}_k; \mu_k) \leq \left\langle \nabla \mathcal{E}_h(\boldsymbol{z}_{k+1}; \mu_k) + \begin{pmatrix} q_{k+1} \\ 0 \end{pmatrix}, \boldsymbol{z}_{k+1} - \boldsymbol{z}_k \right\rangle - D_{\mathcal{E}_h}(\boldsymbol{z}_k, \boldsymbol{z}_{k+1}; \mu_k), \quad (24)$$

where the negative term $-D_{\mathcal{E}_h}(\boldsymbol{z}_k, \boldsymbol{z}_{k+1}; \mu_k)$ is expanded as $-D_h(x_k, x_{k+1}) - \frac{\mu_k}{2}\|y_k - y_{k+1}\|^2$. The inequality is due to the definition of the subgradient.

From Lemma 4.5, we have

$$\left\langle \nabla \mathcal{E}_h(\boldsymbol{z}_{k+1}; \mu_k) + \begin{pmatrix} q_{k+1} \\ 0 \end{pmatrix}, \alpha_k \mathcal{G}(\boldsymbol{z}_{k+1}, \mu_k) \right\rangle \leq -\alpha_k \mathcal{E}(\boldsymbol{z}_{k+1}, \mu_k)$$

$$- \frac{1}{L_k}\|\nabla h(x_{k+1}) + q_{k+1}\|_*^2 - \alpha_k D_h(x^*, x_{k+1}) + \frac{\alpha_k \mu_k}{2}\left(\|x_{k+1} - x^\star\|^2 - \|x_{k+1} - y_{k+1}\|^2\right).$$

The difference between the scheme and the implicit Euler method is

$$\boldsymbol{z}_{k+1} - \boldsymbol{z}_k - \alpha_k \mathcal{G}(\boldsymbol{z}_{k+1}, \mu_k) = \alpha_k \begin{pmatrix} y_k - y_{k+1} + \beta_k(\nabla h(x_{k+1}) - \nabla h(x_k)) \\ 0 \end{pmatrix}.$$

which will bring more terms

$$\langle \nabla_x \mathcal{E}_h(\boldsymbol{z}_{k+1}, \mu_k) + q_{k+1}, \boldsymbol{z}_{k+1} - \boldsymbol{z}_k - \alpha_k \mathcal{G}(\boldsymbol{z}_{k+1}, \mu_k) \rangle$$

$$= \frac{1}{L_k}\left(\nabla h(x_{k+1}) + q_{k+1}, \nabla h(x_{k+1}) - \nabla h(x_k)\right) + \alpha_k \left\langle \nabla h(x_{k+1}) + q_{k+1}, y_k - y_{k+1} \right\rangle.$$

For the first term, we use the identity of squares

$$\frac{1}{L_k}\left(\nabla h(x_{k+1}) + q_{k+1}, \nabla h(x_{k+1}) - \nabla h(x_k)\right)$$

$$= -\frac{1}{2L_k}\|\nabla h(x_k) + q_{k+1}\|_*^2 + \frac{1}{2L_k}\|\nabla h(x_{k+1}) + q_{k+1}\|_*^2 + \frac{1}{2L_k}\|\nabla h(x_{k+1}) - \nabla h(x_k)\|_*^2.$$

As expected, this cross term brings more positive squares but also contribute a negative one.

For the second term, we rewrite as

$$\alpha_k \langle \nabla h(x_{k+1}) + q_{k+1}, y_k - y_{k+1} \rangle = \left\langle \frac{\alpha_k}{\sqrt{\mu_k}} \nabla h(x_{k+1}) + q_{k+1}, \sqrt{\mu_k}(y_k - y_{k+1}) \right\rangle$$

$$= \frac{\alpha_k^2}{2\mu_k} \|\nabla h(x_{k+1}) + q_{k+1}\|_*^2 + \frac{\mu_k}{2} \|y_k - y_{k+1}\|^2 - \frac{1}{2} \left\| \frac{\alpha_k}{\sqrt{\mu_k}}(\nabla h(x_{k+1}) + q_{k+1}) - \sqrt{\mu_k}(y_k - y_{k+1}) \right\|^2$$

$$= \frac{\alpha_k^2}{2\mu_k} \|\nabla h(x_{k+1}) + q_{k+1}\|_*^2 + \frac{\mu_k}{2} \|y_k - y_{k+1}\|^2 - \frac{1}{2} \alpha_k^2 \mu_k \|x_{k+1} - y_{k+1}\|^2.$$

Combining altogether, we get the desired inequality. □

---

**Algorithm 3:** A$^2$GD method for composite optimization

---

**Input:** $x_0, y_0 \in \mathbb{R}^n$, $L_0, \mu_0, R > 0$, tol $> 0$, $\varepsilon > 0$, $m \geq 1$

**1 while** $k = 0$ *or* $\|\nabla f(x_k) + q_k\| > \text{tol}\|\nabla f(x_0)\|$ **do**

**2**    $\alpha_k \leftarrow \sqrt{\mu_k/L_k}$;

**3**    $w_{k+1} \leftarrow \frac{1}{\alpha_k+1} x_k + \frac{\alpha_k}{\alpha_k+1} y_k - \frac{1}{L_k(\alpha_k+1)} \nabla h(x_k)$;

**4**    $x_{k+1} \leftarrow \text{prox}_{\frac{1}{L_k(\alpha_k+1)} g}(w_{k+1})$;

**5**    $q_{k+1} \leftarrow L_k(\alpha_k + 1)(w_{k+1} - x_{k+1})$;

**6**    $y_{k+1} \leftarrow \frac{\alpha_k}{\alpha_k+1} x_{k+1} + \frac{1}{\alpha_k+1} y_k - \frac{\alpha_k}{\mu_k(\alpha_k+1)}(\nabla h(x_{k+1}) + q_{k+1})$;

**7**    $b_k^{(1)} \leftarrow \frac{1}{2L_k}\|\nabla h(x_{k+1}) - \nabla h(x_k)\|^2 - D_h(x_k, x_{k+1})$;

**8**    $b_k^{(2)} \leftarrow -\frac{1}{2L_k}\|\nabla h(x_k) + q_{k+1}\|_*^2 + \frac{\alpha_k \mu_k}{2}\left(R^2 - (1+\alpha_k)\|x_{k+1} - y_{k+1}\|^2\right)$;

**9**    $p_k \leftarrow \frac{1}{1+\alpha_k}(p_{k-1} + b_k^{(1)} + b_k^{(2)})$;

**10**    **if** $p_k > 0$ **then**

**11**      **if** $b_k^{(1)} > 0$ **then**

**12**        $v \leftarrow \frac{2L_k D_f(x_k, x_{k+1})}{\|\nabla f(x_{k+1}) - \nabla f(x_k)\|^2}$, $L_k \leftarrow 3L_k/v$;

**13**      **if** $b_k^{(2)} > 0$ **then**

**14**        $\mu_k \leftarrow \max\left\{\varepsilon, \min\left\{\mu_k, \frac{\|\nabla h(x_k)+q_{k+1}\|^{4/3}}{L_k^{1/3}(R^2-(1+\alpha_k)\|x_{k+1}-y_{k+1}\|^2)^{2/3}}\right\}\right\}$;

**15**      Go to line 2;

**16**    **else**

**17**      $L_k \leftarrow \frac{\|\nabla h(x_{k+1}) - \nabla h(x_k)\|^2}{2D_h(x_k, x_{k+1})}$;

**18**      $\mu_{k+1} \leftarrow \max\left\{\varepsilon, \min\left\{\mu_k, \frac{\|\nabla h(x_k)+q_{k+1}\|^{4/3}}{L_k^{1/3}(R^2-(1+\alpha_k)\|x_{k+1}-y_{k+1}\|^2)^{2/3}}\right\}\right\}$;

**19**    **if** *decay condition* **then**

**20**      $\varepsilon \leftarrow \varepsilon/2$;

**21**      $m \leftarrow \lfloor \sqrt{2} \cdot m \rfloor + 1$;

**22**    $k \leftarrow k + 1$;

---

**Theorem 4.7.** *Let $(x_k, y_k)$ be the iterates generated by Algorithm 3. Assume function $f$ is $\mu$-convex with $\mu \geq 0$. Assume there exists $R > 0$ such that*

$$\|x_k - x^*\| \leq R, \qquad \forall\ k \geq 0.$$

*Let $k_s$ be the total number of steps after halving $\varepsilon$ exactly $s$ times, i.e. $\varepsilon = 2^{-s}\varepsilon_0$.*

*1. When $\mu = 0$, ther exists a constant $C > 0$ so that*

$$\frac{\mathcal{E}_{k_s}}{\mathcal{E}_0} \leq \frac{R^2 + 1}{\left(Ck_s + \varepsilon_0^{-1/2}\right)^2} = \mathcal{O}\left(\frac{1}{k_s^2}\right)$$

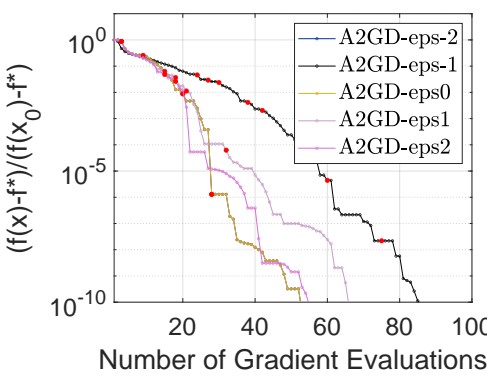 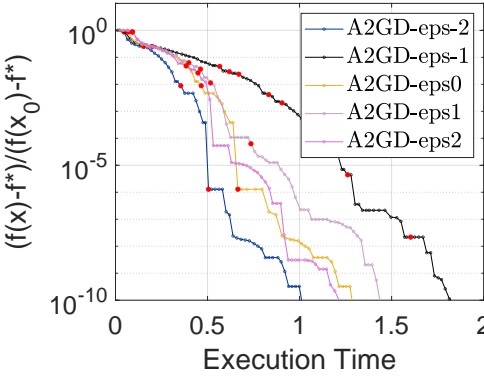

Figure 11: A2GD with different choices of $\varepsilon$, in terms of number of gradients (left) and execution time (right).

2. When $\mu > 0$, the iteration number to achieve $\mathcal{E}_{k_s}/\mathcal{E}_0 \leq (R^2+1)2^{-s}\varepsilon_0 \leq \text{tol}$ is bounded by $\mathcal{O}(\sqrt{L/\mu} \, \ln \text{tol})$,

where $\mathcal{E}_k = \mathcal{E}(\boldsymbol{z}_k; \mu_k) = f(x_k) - f(x^\star) + \frac{\mu_k}{2}\|y_k - x^\star\|^2$.

## APPENDIX D: DISCUSSION

**The Relationship of Running Time with Gradient Evaluations** In the main text, we report convergence primarily in terms of gradient evaluations, which dominate the computational cost for all methods. Consequently, wall-clock time is essentially proportional to the number of gradient (or proximal-gradient) evaluations. To confirm this, we record detailed timings on the composite MLE task, counting each proximal step as one gradient evaluation.

| Method | # Iter | # Grad Eval | Total time | Time/Iter | Grad Time | Grad % |
|--------|--------|-------------|------------|-----------|-----------|--------|
| $A^2GD$ | 2376 | 2382 | 2.71 | $1.14 \times 10^{-3}$ | 1.71 | 63.2% |
| AdProxGD | 20941 | 20941 | 18.68 | $8.92 \times 10^{-4}$ | 14.68 | 78.6% |
| FISTA | 18041 | 18041 | 18.21 | $1.01 \times 10^{-3}$ | 13.80 | 75.8% |
| AOR-HB | 8877 | 8877 | 8.81 | $9.93 \times 10^{-4}$ | 6.70 | 76.1% |

Table 2: Computation cost breakdown on the MLE problem (2).

Gradient (and proximal-gradient) evaluations account for over 60% of the total running time for every method. Although $A^2GD$ incurs slightly higher per-iteration cost due to a few extra vector operations, its much smaller number of gradient evaluations yields nearly a 70% reduction in total time. We also provide error curves versus wall-clock time for the regularized logistic regression on Adult Census Income.

**Ablation Study on the Choice of Hyper-parameter $\varepsilon$** The tolerance $\varepsilon$ is a small positive number that controls $\mu_k$ from below. It is essential in the proof of linear/sub-linear convergence (Theorem 3.2). Numerically, manipulating this parameter will keep the convergence of the algorithm, while making a difference to the convergence rate. This is verified in an ablation study on the regularized logistic regression problem on the Adult Census Income dataset. For $-2 \leq i \leq 2$, the algorithm named "A2GD-eps$i$" means the manual choice $\varepsilon = 1/10^{-6-i}$, where $i = 0$ gives the choice of $\varepsilon$ that we use for all numerical examples in the main context. Figure 11 agrees with the theory, and shows robustness of A2GD with the parameter $\varepsilon$.

**Further Discussion on the Problem Scaling** Apart from the linear example which we discussed in main context, we also tested a range of $\lambda$ values ($\lambda = 10^{-2}, 1, 10^2$) in the

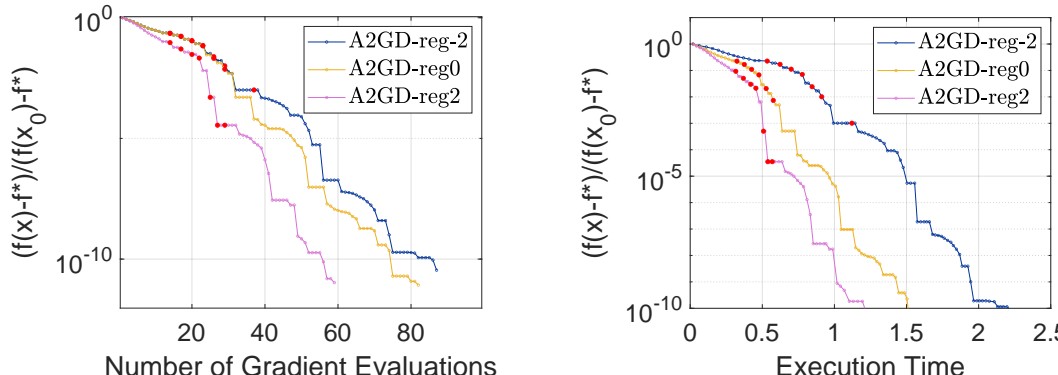

Figure 12: A²GD on regularized logistic regression problem with different regularization constants, in terms of number of gradients (left) and execution time (right).

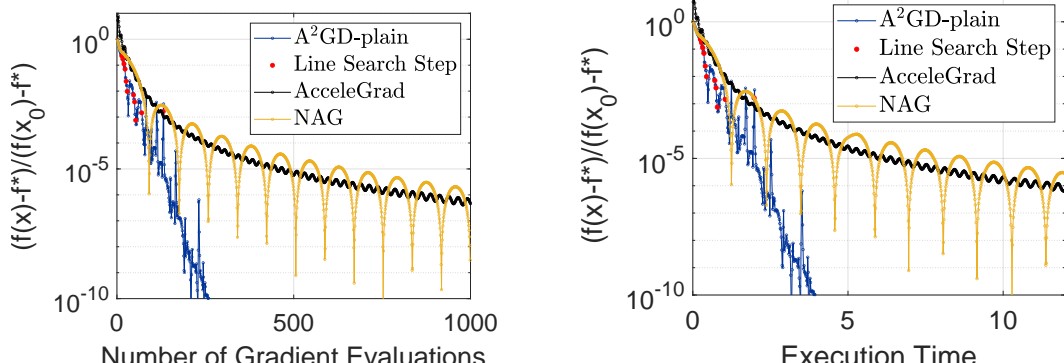

Figure 13: Comparison of methods with adaptive degree of freedom $0, 1, 2$. Results are in terms of number of gradients (left) and execution time (right).

regularized logistic regression problem. In this case, the $\sqrt{\kappa}$ scaling is less apparent because our method performs very well when $\lambda$ is close to 0. Even in the case $\lambda = 10^{-2}$, the adaptive $\mu_k$ does not necessarily remain small during the iterations, which can effectively improve the convergence beyond what the nominal condition number would suggest. Details are in Figure 12.

**An Empirical Study of Adaptive Degree of Freedom**  As mentioned in the Introduction Section, we can classify accelerated gradient methods via the adaptive degree of freedom (ADoF). ADoF-0 simply means non-adaptive accelerated methods, for example NAG. ADoF-1 and ADoF-2 mean the method has 1 and 2 adaptive parameters respectively. For example, the AcceleGrad method proposed in Levy et al. (2018) belongs to ADoF-1, while our A²GD method and a few other baselines in the main context belong to ADoF-2. To see their difference numerically, we compare the 3 methods on the regularized logistic regression problem on Adult Census Income dataset (Figure 13). As neither AcceleGrad nor NAG requires restarting, we also use A²GD-plain for a fair comparison. AcceleGrad performs slightly better than NAG by reducing the oscillations and converging in a reasonable pace, but its improvement is limited due to the semi-adaptivity. In contrast, A²GD outperforms AcceleGrad and NAG dramatically. This can be seen as an example where the additional adaptive parameter (in A²GD, the parameter is $\mu$) brings much improvement.

# APPENDIX E: CONVERGENCE GRAPHS IN TERMS OF EXECUTION TIME

Here, we present the convergence graphs in which the x-axis represents execution time (unit: second). As discussed, for all tested gradient methods, execution time is approximately proportional to the number of gradients, but we still present them here for completeness.

**Regularized Logistic Regression** Below are the results for the regularized logistic regression problem.

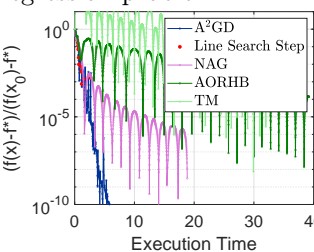
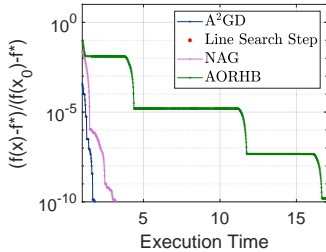
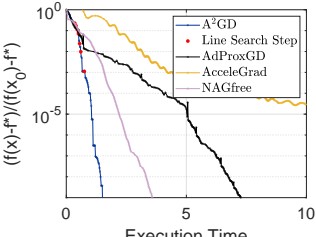

Figure 14: Comparison without restarting.

Figure 15: Comparison with restarting.

Figure 16: $A^2GD$ compared to other adaptive methods.

**Maximum Likelihood Estimation** Below are the results for the maximum likelihood estimation problem.

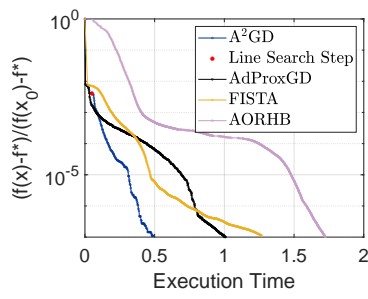
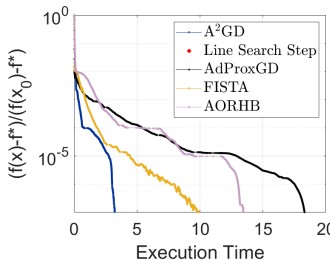

Figure 17: Error curves under setting (1).

Figure 18: Error curves under setting (2).

$\ell_{1-2}$ **Nonconvex Minimization** Below are the results for the $\ell_{1-2}$ nonconvex minimization problem.

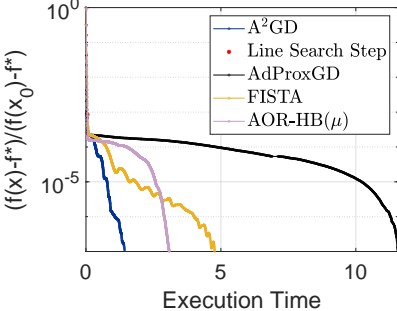

Figure 19: Error curve for $\ell_{1\text{-}2}$ problem with $n = 500, p = 1000$.

## LLM USAGE

In preparing this manuscript, large language models (LLMs) were employed exclusively to assist with language-related tasks, such as improving readability, grammar, and style.

The models were not used for research ideation, development of methods, data analysis, or interpretation of results. All scientific content, including problem formulation, theoretical analysis, and experimental validation, was conceived, executed, and verified entirely by the authors. The authors bear full responsibility for the accuracy and integrity of the manuscript.

## ETHICS STATEMENT

This work is purely theoretical and algorithmic, focusing on convex optimization methods. It does not involve human subjects, sensitive data, or applications that raise ethical concerns related to privacy, security, fairness, or potential harm. All experiments are based on publicly available datasets or synthetic data generated by standard procedures. The authors believe that this work fully adheres to the ICLR Code of Ethics.

## REPRODUCIBILITY STATEMENT

We have taken several measures to ensure the reproducibility of our results. All theoretical assumptions are explicitly stated, and complete proofs are provided in the appendix. For the experimental evaluation, we describe the setup, parameter choices, and baselines in detail in the main text. The source code for our algorithms and experiments are available as supplementary materials. Together, these resources should allow others to reproduce and verify our theoretical and empirical findings.

