# OpenReview forum: "Adaptive Accelerated Gradient Descent Methods for Convex Optimization"
_ICLR.cc/2026/Conference — Submitted to ICLR 2026_

### Official Review · Reviewer_EtSQ · 2025-10-27

**Soundness:** 3
**Presentation:** 3
**Contribution:** 2
**Rating:** 4
**Confidence:** 2

**Summary:**

The article proposes an adaptive (accelerated) gradient descent method that estimates smoothness and strong-convexity parameters on the fly. There are two key ideas: i) designing a suitable energy function that measures descrease, ii) relaxing the decrease condition, such that sufficient decrease is not necessarily imposed at every step, but only in a suitable avergage sense. The ideas are underlined with a few small scale experiments including regularized logisitic regression, a stylized semidefinite programming problem, and a simple nonconvex regularized least-squares problem.

Overall the article is well written and fun to read.

**Strengths:**

The algorithmic approach is relatively simple and the numerical examples (albeit small-scale and idealized) show substantial improvement compared to baselines. The content is presented in a coherent and mostly clear manner.

**Weaknesses:**

- The originality of the article is limited. Adaptive-step size selection is a theme that has been studied since the very early days of optimization (~50ies/60ies). The extension from enforcing decrease of a certain quality-function at every iteration towards enforcing sufficient decrease over multiple iterations is relatively minor.

- In the accelerated situation, which is arguably the less standard one, there are a few algorithmic details that seem rather ad-hoc. These include an ad-hoc lower bound on \mu_k (estimate of strong-convexity), ensuring monotonic descent of function values in accelerated gradient descent (I have some concerns about this, as non-monotonic decrease seems important for achieving acceleration from a theoretical point of view), and a warm-up phase. These ingredients are mentioned quickly, not included in the pseudo-code of Algorithm 1, and is stated that omitting these might cause instability. Hence, the effect of these ad-hoc ingredients seems important and should therefore be mentioned in the pseudo-code and corresponding ablation studies should be made.

- The experiments are rather limited and include relatively stylized problems (although three fundamentally different problem types). In particular, acceleration is claimed and for this sort of article I would expect to have seen numerical results that indicate how the convergence rate scales with O(\sqr(kappa) log(1/eps)) in the strongly convex regime, i.e., studies with varying kappa and empirically fitting the corresponding convergence rate, and results that show how O(1/k^2) is achieved in a non-strongly-convex setting. Some of this weakness could be addressed if the author could run their algorithm e.g. on problem (14) with varying \lambda, and even for \lambda=0 and using the same hyper-parameters in their algorithm for all problem instances.

**Questions:**

Is there a direct relationship between number of gradient evaluations and execution time? If so, I would recommend the authors to point this out. It seems that additional gradient evaluations are needed to carry out the line search, since the error criteria depends on successive iterates. Are these gradient evaluations accounted for in the numerical results?

Would the method also work in the context of stochastic gradient descent/mini-batching?

---

> ### Author Response · Authors · 2025-11-17
>
> We thank the reviewer for the thorough comments and constructive suggestions. Below, we address each concern in turn. We have updated our submission accordingly.
>
> &nbsp;
>
> ## **Originality**
>
> > The originality of the article is limited ... enforcing sufficient decrease over multiple iterations is relatively minor.
>
> We thank the reviewer for the comment, but we respectfully believe the contribution goes well beyond a minor extension of classical adaptive step-size ideas. Classical adaptive rules from the 1950s–60s (e.g., Armijo–Goldstein) apply only to non-accelerated gradient methods; they are not suitable for accelerated schemes.
>
> As the reviewer notes, **adaptive accelerated** methods are rare. Existing approaches typically require explicit smoothness estimates, impose restrictive structural assumptions, or lose accelerated linear rates. Our framework is, to our knowledge, the first to obtain accelerated linear rates using only local information, without relying on a priori smoothness or convexity parameters.
>
> Most notably, we combine Lyapunov analysis for dynamical systems with adaptivity, using an accumulation argument (rather than line search at every step) and a parameter-update structure that has no analogue in classical methods.
>
> Thus, the contribution is not simply relaxing a per-iteration descent rule, but developing a new analysis framework that enables **provably adaptive acceleration**. The numerical results further show **substantially stronger performance** than existing methods, including those that incorporate both adaptivity and acceleration.
>
> &nbsp;
>
> ## **Ablation study**
>
> > In the accelerated situation, which is arguably the less standard one, there are a few algorithmic details that seem rather ad-hoc. ... Hence, the effect of these ad-hoc ingredients seems important and should therefore be mentioned in the pseudo-code and corresponding ablation studies should be made.
>
> We agree that a careful, detail-level comparison is important.  To isolate the source of improvement, we conducted a set of ablation studies in our revision:
>
> **(1) Warm-up phase (10 GD iterations).**
> We removed the warm-up and manually selected $(\mu_0, R, \varepsilon)$. The performance of A2GD remains strong even when these parameters vary by a factor $10^3$, indicating that the warm-up is not essential for good performance.
>
> **(2) Accept/reject rule and restart.**
> We removed the accept/reject rule and restart strategy. The plain A2GD exhibits some oscillations but still outperforms other *plain* accelerated baselines. The restart mechanism, therefore, is not necessary but does help reduce oscillations.
>
> **(3) Fair comparison with restarts.**
> For fairness, we added and fine-tuned restart and accept/reject policies to NAG and other baselines (if possible). While this significantly improves their convergence, A2GD still performs best and requires roughly half the number of iterations of NAG with restart.
>
> Overall, these ablations show that the adaptivity inherent in A2GD, not warm-up or auxiliary components, is the key driver of the improved convergence.
>
> &nbsp;
>
> ## **Scaling behavior**
>
> > In particular, acceleration is claimed and for this sort of article I would expect to have seen numerical results that indicate how the convergence rate scales with O(\sqr(kappa) log(1/eps)) in the strongly convex regime, i.e., studies with varying kappa and empirically fitting the corresponding convergence rate, and results that show how O(1/k^2) is achieved in a non-strongly-convex setting.
>
> We include the linear (quadratic) case from the finite element discretization of $-\Delta u = f$ on a mesh $\mathcal T_h$. When the condition number increases from $\kappa$ to $4\kappa$, the number of gradient steps grows by roughly a factor of $2$, while the wall-clock time increases by about a factor of $8$ due to the fourfold increase in problem size. This behavior is consistent with the method’s $\sqrt{\kappa}$ convergence rate.
>
> We also test different tolerances $\varepsilon$ and observe the predicted scaling; see the updated Appendix (to be uploaded) for details.
>
> |$h$|$n$|$\kappa$|#Grad|Time(s)|
> |---|---|--------|-----|-------|
> |1/20|1262|$7.85\times10^2$|184|0.01|
> |1/40|5167|$3.15\times10^3$|303|0.07|
> |1/80|20908|$1.30\times10^4$|607|0.56|
> |1/160|84120|$5.32\times10^4$|987|4.52|
>
> > Some of this weakness could be addressed if the author could run their algorithm e.g. on problem (14) with varying \lambda, and even for \lambda=0 and using the same hyper-parameters in their algorithm for all problem instances.
>
> We also tested a range of $\lambda$ values ($\lambda=10^{-2}, 1, 10^2$) in the regression problem. In this case, the $\sqrt{\kappa}$ scaling is less apparent because our method performs very well when $\lambda$ is close to $0$. Even in the case $\mu = 10^{-2}$, the adaptive $\mu_k$ does not necessarily remain small during the iterations, which can effectively improve the convergence beyond what the condition number would suggest.

---

> ### Author Response · Authors · 2025-11-17
>
> &nbsp;
>
> ## **Computational time**
>
>
> > Is there a direct relationship between number of gradient evaluations and execution time? If so, I would recommend the authors to point this out. It seems that additional gradient evaluations are needed to carry out the line search, since the error criteria depends on successive iterates. Are these gradient evaluations accounted for in the numerical results?
>
> Thank you for the comment. In the revision, we will include plots of function value versus wall-clock time for all examples. We do include additional gradient evaluation when line search is triggered.
>
> In practice, gradient evaluations dominate the total cost for all methods. For the composite MLE task, the proximal step requires an eigen-decomposition, which we count as part of a “gradient evaluation,” since the prox is effectively an implicit gradient call. Below is the cost breakdown we observed:
>
> - Gradient evaluations (including prox/eigen-decomp) account for over 60% of total runtime for every method.
> - A2GD incurs a slightly higher per-iteration cost due to a few extra vector operations, but
> - **A2GD reduces overall runtime by about 70%** because it cuts roughly **75% of gradient evaluations**.
>
> We have added this breakdown and the corresponding timing plots in the revised version.
>
> | Method   | # Iterations | # Gradients | Total time | Time per iter | Gradient Time | Gradient Proportion |
> | -------- | ------------ | ----------- | ---------- | ------------- | ------------- | ------------------- |
> | A2GD     | 2376         | 2382        | 2.71       | 1.14e-3       | 1.71          | 63.2%               |
> | AdProxGD | 20941        | 20941       | 18.68      | 8.92e-4       | 14.68         | 78.6%               |
> | FISTA    | 18041        | 18041       | 18.21      | 1.01e-3       | 13.80         | 75.8%               |
> | AOR-HB   | 8877         | 8877        | 8.81       | 9.92e-4       | 6.70          | 76.1%               |
>
> Gradient evaluation (including prox calls) account for >60% for all methods. The running time per iteration of A2GD is slightly more than other methods because of the few additional vector operations. Overall, A2GD reduces the running time by ~70% by cutting ~75% of gradient steps.
>
> &nbsp;
>
> ## **Stochastic extension**
>
> > Would the method also work in the context of stochastic gradient descent/mini-batching?
>
> Unfortunately, developing a fully adaptive theory for momentum methods in the stochastic setting remains an open challenge. For example, in *Online Adaptive Methods, Universality and Acceleration* (NeurIPS 2018), the accelerated $O(1/k^2)$ rate can be proved in the deterministic case, but only the standard $O(1/\sqrt{k})$ rate is achievable under stochastic gradients.
>
> A key difficulty is that SGD analyses rely on explicit variance control, which interacts nontrivially with adaptive step sizes. In stochastic optimization, **variance reduction** becomes the central issue, and the momentum terms in methods such as Adam can be interpreted from this variance-reduction perspective.

---

> ### Comment · Reviewer_EtSQ · 2025-11-27
> **Rebuttal Acknowledgement**
>
> Dear authors,
>
> I would like to thank you for the time and effort spent in addressing the issues I mentioned. The points were addressed in a satisfactory manner, the manuscript has been substantially strengthened, and I will increase my score accordingly. Nonetheless, I still believe that the contributions on a fundamental level are rather incremental. The authors might consider releasing their algorithm as an open-source software package, since it might be useful for others and/or target a specific application domain, where their method will shine.

---

> > ### Author Response · Authors · 2025-11-27
> >
> > Thank you for the positive feedback and for increasing the score. We appreciate the suggestion to release the algorithm as an open-source software package. To comply with the double-blind policy, we currently provide an anonymized implementation in the supplementary materials to support reproducibility. After the review process concludes, we plan to release a full open-source package with hands-on documentation and examples, so that all results in the paper can be easily reproduced and the method can be readily used by the community.

---

### Official Review · Reviewer_4Q9j · 2025-10-29

**Soundness:** 3
**Presentation:** 3
**Contribution:** 2
**Rating:** 4
**Confidence:** 3

**Summary:**

The paper studies gradient descent methods with line search, aiming to reduce the number of line search steps. In the standard method, line search is performed in every iteration where the error term is positive. In the new method, an accumulation of additive error is maintained and line search is only performed when the sum of error terms is positive. As long as the sum of error terms is negative, the objective value goes down exponentially with the observed step sizes.

**Strengths:**

The proposed method is relatively simple and exploits a natural observation that there are several error terms in the standard analysis and forcing all terms to be negative might be too conservative and costly. The proofs build on largely the standard arguments with one new argument for keeping track of the sum of the error terms.

The experiments show strong performance compared with other theoretical adaptive methods and un-tuned non-adaptive methods.

**Weaknesses:**

The accelerated result has two new hyperparameters R and epsilon compared with the standard method, which negates the adaptivity advantage of the method.

There are a lot of heuristics in the new method (heuristic line search, warm-start to set mu) and several extra hyperparameters. In the experimental comparison, it looks like the non adaptive methods are not tuned at all, leading to oscillating behavior. This is already observed in the literature e.g. O'Donoghue, Candes. Adaptive Restart for Accelerated Gradient Schemes FoCM 2013. This is a very pessimistic view and not aligned with any practical deployment of these methods. What if we also run these methods briefly/on smaller data and tune the step sizes (or restart) to avoid the oscillation?

Another weakness is that it is unclear how the new method based on line search is going to generalize to the stochastic setting. The introduction of the paper suggests that an important motivation is to understand and improve upon Adam, AMSgrad, etc. These methods are generally applied in the stochastic setting and do not use line search at all. The experimental comparisons also do not involve any of these methods.

Other comments: in figures 5,6,7, the red dots are not visible.

**Questions:**

Could you please comment on the comparison between this work and the works on adaptive algorithms without line search, for both deterministic and stochastic settings such as
Online Adaptive Methods, Universality and Acceleration. NeurIPS 18.

---

> ### Author Response · Authors · 2025-11-17
>
> We thank the reviewer for the thorough comments and constructive suggestions. Below, we address each concern in turn. We have updated our submission accordingly.
>
> &nbsp;
>
> ## **Additional heuristics and hyperparameters**
>
> > There are a lot of heuristics in the new method (heuristic line search, warm-start to set mu) and several extra hyperparameters. In the experimental comparison, it looks like the non adaptive methods are not tuned at all, leading to oscillating behavior. ... This is a very pessimistic view and not aligned with any practical deployment of these methods. What if we also run these methods briefly/on smaller data and tune the step sizes (or restart) to avoid the oscillation?
>
> We agree that a careful, detail-level comparison is important.  To isolate the source of improvement, we have conducted a set of ablation studies in our revision:
>
> **(1) Warm-up phase (10 GD iterations).**
> We removed the warm-up and manually selected $(\mu_0, R, \varepsilon)$. The performance of A2GD remains strong even when these parameters vary by a factor $10^3$, indicating that the warm-up is not essential for good performance.
>
> **(2) Accept/reject rule and restart.**
> We removed the accept/reject rule and restart strategy. The plain A2GD exhibits some oscillations but still outperforms other *plain* accelerated baselines. The restart mechanism, therefore, is not necessary but does help reduce oscillations.
>
> **(3) Fair comparison with restarts.**
> For fairness, we added the same restart and accept/reject policy to NAG and other baselines (if possible). While this significantly improves their convergence, A2GD still performs best and requires roughly half the number of iterations of NAG with restart.
>
> Overall, these ablations show that the adaptivity inherent in A2GD, instead of warm-up or auxiliary components, is the key driver of the improved convergence.
>
> &nbsp;
>
> ## **Stochastic extension**
>
> > Another weakness is that it is unclear how the new method based on line search is going to generalize to the stochastic setting. The introduction of the paper suggests that an important motivation is to understand and improve upon Adam, AMSgrad, etc. These methods are generally applied in the stochastic setting and do not use line search at all. The experimental comparisons also do not involve any of these methods.
>
> Unfortunately, developing a fully adaptive theory for momentum methods in the stochastic setting remains an open challenge. For example, in *Online Adaptive Methods, Universality and Acceleration* (NeurIPS 2018), the accelerated $O(1/k^2)$ rate can be proved in the deterministic case, but only the standard $O(1/\sqrt{k})$ rate is achievable under stochastic gradients.
>
> A key difficulty is that SGD analyses rely on explicit variance control, which interacts nontrivially with adaptive step sizes. In stochastic optimization, **variance reduction** becomes the central issue, and the momentum terms in methods such as Adam can be interpreted from this variance-reduction perspective.
>
> &nbsp;
>
> ## **Comparison with AcceleGrad**
>
> > Could you please comment on the comparison between this work and the works on adaptive algorithms without line search, for both deterministic and stochastic settings such as Online Adaptive Methods, Universality and Acceleration. NeurIPS 18.
>
> In our numerical examples, we have compared against line-search-free methods such as AdProxGD (Malitsky & Mishchenko, 2024) and NAGfree (Cavalcanti et al., 2025).
>
> Thank you for pointing us to *Online Adaptive Methods, Universality and Acceleration* (NeurIPS 2018); we apologize for overlooking it in our literature review. We agree that the AcceleGrad method is closely related, but our approach differs in several key aspects:
>
> **(1) Two adaptive parameters vs. one.**
> AcceleGrad adapts only the smoothness $L_k$ through the step size. Our method adapts **both** $L_k$ and the local convexity $\mu_k$, the latter being much harder to estimate. We address this using our accumulated Lyapunov framework, which has no analogue in AcceleGrad.
>
> **(2) Stronger empirical performance.**
> This extra degree of freedom yields noticeably better numerical behavior. We will include a direct comparison of regularized logistic regression (Adult Census dataset). Even without restart or accept–reject rules (A2GD-plain), our method outperforms AcceleGrad and NAG.
>
> **(3) Stochastic setting.**
> AcceleGrad extends to stochastic gradients but achieves only the standard $O(1/\sqrt{k})$ rate. As noted earlier, obtaining *accelerated* stochastic rates is substantially harder; extending A2GD to this setting is an interesting direction for future work.
>
> We have added a dedicated comparison with AcceleGrad in the revised version, specifying the differences.
>
> &nbsp;
>
> ## **Other Questions**
>
> > Other comments: in figures 5,6,7, the red dots are not visible.
>
> Thank you for pointing this out. We will make them more visible by increasing the size of the dots.

---

> > ### Comment · Reviewer_4Q9j · 2025-11-27
> > **Acknowledgment**
> >
> > I thank the authors for the responses. The new experiments addressed my concern regarding the contributions of various heuristics including restarts. I continue to think that theoretical contribution is limited due to the lack of connections to the stochastic settings, whereas many previous methods can at least address the stochastic setting without acceleration. Nonetheless, the method seems to perform well compared with other theoretical methods in experiments and can provide an additional tool for implementations of future methods. I increased my score.

---

> > > ### Author Response · Authors · 2025-11-27
> > >
> > > We thank the reviewer for the positive feedback and for increasing the score. We acknowledge the limitation of our current analysis in the stochastic setting, and we agree this is an interesting direction for future work. We are glad that the additional experiments clarified the effect of the heuristics and that the method may be useful for future implementations.

---

### Official Review · Reviewer_1uuQ · 2025-10-31

**Soundness:** 3
**Presentation:** 3
**Contribution:** 3
**Rating:** 6
**Confidence:** 3

**Summary:**

This paper presents a "line search-reduced" adaptive acceleration scheme for convex and composite optimization. The key idea is to track an accumulated perturbation variable and trigger line search only when it becomes positive. This mechanism adaptively updates both the smoothness and strong-convexity estimates, achieving $O(1/k^2)$ for convex problems and accelerated linear convergence for strongly convex ones.

**Strengths:**

- The paper proposes an interesting "line search-reduced" adaptive acceleration method through the elegant use of a running perturbation balance $p_k$ to trigger line search only when the Lyapunov-stability condition $p_k \leq 0$ is violated.

- The paper provides clear convergence guarantees for both convex and strongly convex objectives. The way $L_k$ and $\mu_k$ are adaptively controlled makes sense and the Lyapunov-based analysis is well presented.

- It's nice to see a clean algorithmic idea that really tries to reduce line search overhead while achieving provable acceleration. That said, it'd be even more convincing to see how sensitive the method is to the various heuristics and hyperparameters (see Weaknesses).

**Weaknesses:**

- There's no theoretical upper bound on the number of line search activations. While each backtracking loop is shown to finish in $O(\log L)$ steps, the paper doesn't say how often these activations happen overall. In the worst case, frequent triggers could blow up the total gradient evaluations and weaken the claimed overall complexity.

- The experimental evaluation is somewhat limited. Results are shown only in terms of gradient evaluations; no wall-clock time, memory or cost breakdown (e.g., line-search triggers, eigen-decomp/prox calls). Especially for the composite MLE task, where each prox involves an eigen-decomp, the claimed runtime savings aren't quantified. Also, all tests are mid-scale convex problems.

- The method uses several heuristics, such as the accept/reject rule, restart, AdProxGD warm-up, but the paper doesn't analyze how much these actually matter.

**Questions:**

- Can you provide a theoretical or empirical upper bound on the number of line search activations? Is the frequency roughly sublinear or constant in practice?

- Could you report wall-clock time or a cost breakdown (gradient, prox, line search) to show real runtime benefit?

- How does the proposed method scale to larger problems? How does it perform on ill-conditioned problems, e.g., by varying $L/\mu$ to test its robustness?

- Can you share ablation results for the heuristics (accept/reject rule, restarts, warm-up) to see how much they impact convergence and stability?

- Have you tested parameter sensitivity ($\mu_0, L_0, R, \varepsilon$, etc.) to check robustness of the adaptive behavior?

---

> ### Author Response · Authors · 2025-11-16
>
> We thank the reviewer for recognizing our contributions and for the thorough, constructive feedback. Below, we address each concern in turn and we have updated our submission accordingly.
>
> &nbsp;
>
> ## **Number of linear search activations**
> > Can you provide a theoretical or empirical upper bound on the number of line search activations?
>
> We currently do not have a theoretical upper bound on the *total* number of line-search activations beyond the trivial $O(\log L)$ bound for each individual backtracking loop. This is difficult to strengthen due to the variability of local curvature in general convex functions, which can lead to irregular triggering patterns.
>
> Empirically, however, line search is activated only a small number of times—typically fewer than $10$ over the entire run—and most activations occur early in the optimization to secure sufficient initial decay. In the revision, we will include a summary table reporting activation counts across different portions of the run (first 25%, 25–50%, last 50%), showing that the frequency is essentially constant and does not impact overall complexity.
>
> |Problem|Iter|0–25%|25–50%|50–100%|Total|
> |-------|----|-----|------|--------|-----|
> |Reg. Logistic Regression|83|3|4|0|7|
> |MLE $\kappa=10^2$|185|3|0|0|3|
> |MLE $\kappa=10^4$|2376|6|0|0|6|
> |$\ell_{1-2}$ Nonconvex|693|6|0|0|6|
>
> &nbsp;
>
> ## **Computational time**
>
> > Could you report wall-clock time or a cost breakdown (gradient, prox, line search) to show real runtime benefit?
>
> In the revision, we include plots of function value versus wall-clock time for all examples. Since none of the methods store past iterates, their memory usage is comparable, scaling as $O(d)$ for problem dimension $d$.
>
> In practice, gradient evaluations dominate the total cost for all methods. For the composite MLE task, the proximal step requires an eigen-decomposition, which we count as part of a “gradient evaluation,” since the prox is effectively an implicit gradient call. Below is the cost breakdown we observed:
>
> - Gradient evaluations (including prox/eigen-decomp) account for over 60% of total runtime for every method.
> - A2GD incurs a slightly higher per-iteration cost due to a few extra vector operations, but
> - **A2GD reduces overall runtime by about 70%** because it cuts roughly **75% of gradient evaluations**.
>
> We have add this breakdown and the corresponding timing plots in the revised version.
>
> |Mthd|Iter|Grad|Time|T/it|GTime|G%|
> |----|----|----|----|-----|------|----|
> |A2GD|2376|2382|2.71|1.14e-3|1.71|63.2%|
> |AdPGD|20941|20941|18.68|8.92e-4|14.68|78.6%|
> |FISTA|18041|18041|18.21|1.01e-3|13.80|75.8%|
> |AOR-HB|8877|8877|8.81|9.93e-4|6.70|76.1%|
>
> &nbsp;
>
> ## **Isolate the source of improvement**
>
> > Can you share ablation results for the heuristics (accept/reject rule, restarts, warm-up) to see how much they impact convergence and stability?
>
>  To isolate the source of improvement, we have conducted a set of ablation studies:
>
> **(1) Warm-up phase (10 GD iterations).**
> We removed the warm-up and manually selected \((\mu_0, R, \varepsilon)\). The performance of A2GD remains strong even when these parameters vary by a factor of  \(1000\), indicating that the warm-up is not essential for good performance.
>
> **(2) Accept/reject rule and restart.**
> We removed the accept/reject rule and restart strategy. The plain A2GD exhibits some oscillations but still outperforms other *plain* accelerated baselines. The restart mechanism, therefore, is not necessary but does help reduce oscillations.
>
> **(3) Fair comparison with restarts.**
> For fairness, we added the same restart and accept/reject policy to NAG and other baselines (if possible). While this significantly improves their convergence, A2GD still performs best and requires roughly half the number of iterations of NAG with restart.
>
> Overall, these ablations show that the adaptivity inherent in A2GD, instead of warm-up or auxiliary components, is the key driver of the improved convergence.
>
> &nbsp;
>
> ## **Scaling behavior**
>
> > How does the proposed method scale to larger problems? How does it perform on ill-conditioned problems, e.g., by varying $L/\mu$ to test its robustness?
>
> We include the linear (quadratic) case from the finite element discretization of $-\Delta u = f$ on a mesh $\mathcal T_h$. When the condition number increases from $\kappa$ to $4\kappa$, the number of gradient steps grows by roughly a factor of $2$, while the wall-clock time increases by about a factor of $8$ due to the fourfold increase in problem size. This behavior is consistent with the method’s $\sqrt{\kappa}$ convergence rate.
>
> |$h$|$n$|$\kappa$|#Grad|Time|
> |---|---|--------|----------|---------|
> |1/20|1262|7.85e2|184|0.01|
> |1/40|5167|3.15e3|303|0.07|
> |1/80|20908|1.30e4|607|0.56|
> |1/160|84120|5.32e4|987|4.52|

---

### Official Review · Reviewer_8qUb · 2025-11-05

**Soundness:** 2
**Presentation:** 3
**Contribution:** 2
**Rating:** 4
**Confidence:** 3

**Summary:**

This paper proposes a new line-search-based accelerated gradient method for smooth (strongly) convex minimization. Traditional line-search-based methods typically require multiple function evaluations per iteration, whereas the proposed Lyapunov-based approach triggers line search only when the accumulated perturbation becomes positive. The analysis shows that the method achieves a near-optimal convergence rate while significantly reducing the number of line-search operations. Experimental results demonstrate that the proposed method outperforms existing line-search-based and adaptive accelerated methods.

**Strengths:**

Because line search can be computationally expensive, there has been considerable interest in developing line-search-free adaptive methods. In this context, proposing a method that triggers line search only occasionally, when it is truly needed, is novel and worth investigating.

**Weaknesses:**

Although I found the analysis and the results in Figures 1 and 2 for the gradient descent interesting, the proposed Algorithm 1 introduces several additional components, such as initialization with ten iterations of gradient descent and restarting, which make it difficult to isolate the source of improvement, especially in the experimental results. In other words, I would like to see a fair comparison focusing solely on line-search-free aspect, without incorporating other auxiliary components. For example, all existing methods in Figure 3 exhibit oscillatory behavior, and it is well known that restarting alone can substantially mitigate such oscillations, as observed in your method.

**Questions:**

- Line 207: Is it correct that an approximation $\mu_k=\min_{1\le i\le k} L_k$ is used to compute $\delta_k$? This choice seems like a potential overestimation, especially since $L_k$ is rarely updated. Are you plotting $p_k$ in Figure 1 using this $\mu_k$? If yes, how would the results differ if the exact $\mu$ were used in computing $p_k$?

- Line 220: I am not sure what is meant by "in a weighed $\ell_2$ sense". What exactly is the weight in this context?

- Line 253: Could you clarify what is meant by "improving efficiency"? It appears that, although line search is not triggered, the method still employs the adaptive step size $\alpha_{k+1}$, which is commonly used in line-search-free approaches. If this is correct, I believe the paper's claimed contribution should be reconsidered. When $\alpha_{k+1}$ is already adaptively chosen, it is not clear why line search remains necessary.

- Line 257: Do you also use this approximation when computing $b_k^{(1)}$?

- Line 273: Could you explain the motivation for considering the non-standard Hessian-based Nesterov accelerated gradient in (11) ?

---

> ### Author Response · Authors · 2025-11-16
>
> We thank the reviewer for the thorough comments and constructive suggestions. Below, we address each concern in turn, and we have updated our submission accordingly.
>
> &nbsp;
>
> ## 1. **Isolate the source of improvement**
>
> > I would like to see a fair comparison focusing solely on the line-search-free aspect, without incorporating other auxiliary components.
>
> Thank you for the thoughtful comments. To isolate the source of improvement, we have conducted a set of ablation studies:
>
> **(1) Warm-up phase (10 GD iterations).**
> We removed the warm-up and manually selected $(\mu_0, R, \varepsilon)$. The performance of A2GD remains strong even when these parameters vary by a factor $10^3$, indicating that the warm-up is not essential for good performance.
>
> **(2) Accept/reject rule and restart.**
> We removed the accept/reject rule and restart strategy. The plain A2GD exhibits some oscillations but still outperforms other *plain* accelerated baselines. The restart mechanism, therefore, is not necessary but does help reduce oscillations.
>
> **(3) Fair comparison with restarts.**
> For fairness, we added and fine-tuned restart and accept/reject policy to NAG and other baselines (if possible). While this significantly improves their convergence, A2GD still performs best and requires roughly half the number of iterations of NAG with restart.
>
> Overall, these ablations show that the adaptivity inherent in A2GD, instead of warm-up or auxiliary components, is the key driver of the improved convergence.
>
> &nbsp;
>
> ## 2. **Estimate of convexity constant**
>
> > Line 207: Is it correct that an approximation $\mu_k=\min_{1\leq i\leq L_k} L_k$ is used to compute $\mu_k$? This choice seems like a potential overestimation, especially since $L_k$ is rarely updated. Are you plotting $p_k$ in Figure 1 using this $\mu_k$? If yes, how would the results differ if the exact $\mu$ were used in computing $p_k$?
>
> In Figure 1, we plot $p_k$ using the true $\mu$, as the figure is purely illustrative. To avoid confusion, we will remove the sentence “$\mu_k=\min_{1\le i\le L_k} L_i$” from Section 2.
>
> In the A2GD algorithm, however, we use $\mu_k = \min_{1\le i\le k} L_i$. We agree that this may initially overestimate the true $\mu$, especially in ill-conditioned problems. However, $\mu_k$ decreases rapidly during the first few A2GD iterations due to the adaptivity mechanism, so the initial overestimation has little impact. This is consistent with our ablation study (added to the numerical section): even when $\mu_0$ varies by a factor of $1000$, the performance remains stable.
>
> &nbsp;
>
> ## 3. **Adaptive step size and line search**
>
>  > Line 253: Could you clarify what is meant by "improving efficiency"? It appears that, although line search is not triggered, the method still employs the adaptive step size $\alpha_{k+1}$, which is commonly used in line-search-free approaches. If this is correct, I believe the paper's claimed contribution should be reconsidered. When $\alpha_{k+1}$ is already adaptively chosen, it is not clear why line search remains necessary.
>
> We adaptively update the step size and use it as the initial guess for the next iteration. This moves the iterates closer to the region where $p_k \le 0$ holds, reducing the likelihood of triggering line search and thereby improving efficiency.
>
> However, using this adaptive step size alone does not guarantee convergence (as seen in methods such as Barzilai–Borwein). The line search is still necessary to enforce $p_k \le 0$, which is crucial for the theoretical convergence of A2GD.
>
> &nbsp;
>
> ## **Other Questions**
>
>
> > Line 220: I am not sure what is meant by "in a weighed $\ell_2$ sense". What exactly is the weight in this context?
>
> Here $p_k = \sum_{i=0}^k c_i  b_i$ is a weighted sum with weights $c_i = \prod_{j=i}^k \delta_j$, which corresponds to an $\ell_1$–type weighted norm, not $\ell_2$. We have corrected this typo in the updated version. Thank you for pointing it out.
>
> > Line 257: Do you also use this approximation when computing $b_k^{(1)}$?
>
> Yes, we use this approximation to compute $b_k^{(1)}$ in our implementations.
>
> > Line 273: Could you explain the motivation for considering the non-standard Hessian-based Nesterov accelerated gradient in (11) ?
>
> We use the Hessian-based Nesterov Gradient (HNAG) formulation because it keeps the error dynamics algebraically simple. In particular, it allows us to derive an exact Lyapunov difference identity (Lemma 3.1). By contrast, the standard Nesterov analysis relies on estimate sequences, where the influence of adaptive $\mu_k$ and $L_k$ becomes opaque. The HNAG formulation makes these dependencies explicit and enables a cleaner and more transparent analysis.

---

### Author Response · Authors · 2025-11-20
**Summary of Key Revisions**

We have uploaded a new PDF file. Across all reviewer responses, the revision incorporates the following major improvements:

1. **Comprehensive Ablation Studies**
   We include detailed ablations isolating the effects of warm-up, restart, accept/reject rules, and parameter choices. These confirm that the **core adaptivity of A2GD**—not auxiliary heuristics—is responsible for the performance gains. Fair comparisons with restart-enabled baselines are also added.

2. **Wall-Clock Time Evaluation**
   In addition to gradient-based convergence plots, we now report convergence versus wall-clock time for the scaling experiment in Appendix D and E. Since gradient evaluation dominates the computational cost, the timing-based error curves closely match the gradient-step plots. In the main text, gradient-based convergence remains the primary metric, with extra gradient evaluations from line search explicitly marked red.

3. **Enhanced Scaling and Robustness Tests**
   We add experiments demonstrating:
   - **Condition-number scaling** is consistent with the expected $\sqrt{\kappa}$ behavior,
   - **Robustness** to variations in $\mu_0$, $L_0$, $R$, and $\varepsilon$,
   - **Line-search activation** is lessn than **10** triggers in all cases.

These updates directly address reviewer concerns regarding fairness, empirical support, and robustness. We thank all reviewers for their careful reading and constructive feedback.

In the revised PDF, we **highlight our changes in red**. For the newly added example and appendix materials, we **highlight only the section titles** to avoid clutter while making the additions easy to locate.

---

> ### Author Response · Authors · 2025-12-01
>
> We thank all reviewers again for their careful evaluations and constructive feedback. We especially appreciate reviewers **4Q9j** and **EtSQ** for their positive acknowledgement and for updating their scores accordingly.
>
> We also thank reviewer **EtSQ** for the suggestion to release the algorithm as an open-source software package. After the double-blind review period ends, we will upload the code to GitHub so the community can easily reproduce and build upon our work.
>
> Extension to non-convex and stochastic settings is an important direction for future research, and we appreciate the reviewers’ comments highlighting these direction.

---

### Meta-Review · Area_Chair_EPq8 · 2026-01-17

**Summary:**

This paper proposes an accelerated and adaptive algorithm A$^2$GD for solving additively composite problem $\min_x h(x) + g(x)$ where $h$ is L smooth and $g$ is convex. The authors prove the rate $O(1/k^2)$ when the objective is not strongly convex and an exponential rate leading to a complexity $O(\sqrt{L/\mu}\epsilon)$ for getting $f(x) - f(x^\star) \leq \epsilon$. The authors conduct many practical experiments to show the promise of their method, against both nonaccelerated adaptive methods such as the method of Malitsky and Mishchenko and accelerated nonadaptive methods such as FISTA. The authors also added comparisons with Accelegrad in their revision.

Even though the practical performance of the method seems indeed promising, the mismatch between the theory and the practice is rather problematic. Indeed, it is not clear what the "decay condition" is in practice and theory. The theorem statement talks about a constant $R$ that is supposed to bound $\|x_k-x^\star\|$ which is only put as a tuning parameter in the experiments. Indeed, for many methods, such as Malitsky and Mishchenko, this is a result that is proven rather than assumed. It is also rather confusing that most of these assumptions are not discussed properly and explicitly in the text. One disadvantage compared to Accelegrad is that the current work cannot be extended to the stochastic case whereas Accelegrad gets an optimal rate in that case too. On the other hand, Accelegrad does not adapt to the strong convexity parameter.

**Reviewer Concerns:**

Reviewers asked the authors to characterize how much each of the "bells and whistles" in the algorithm mattered and the authors included ablation studies to address. Reviewers also asked about comparison to Accelegrad from the work of Levy et al and the authors stated that this work cannot adapt to strong convexity and they also showed that Accelegrad does not perform better than A2GD in practice.

**Reviewer Scores:**

Reviewer EtSQ and 4Q9j would increase their score if a full discussion took place. On the other hand, I share the concerns of 4Q9j regarding the weakness of the theoretical results and I remain unsure whether the theoretical results are on the level of a top venue.

Reviewer 1uuQ would keep their accept recommendation.

Reviewer 8qUb would probably increase their score to a 6 as well.

---

### Decision · Program_Chairs · 2026-01-26

Reject